# Monoubiquitination of ASXLs controls the deubiquitinase activity of the tumor suppressor BAP1

Salima Daou[1,2], Haithem Barbour[1], Oumaima Ahmed[1], Louis Masclef [1], Caroline Baril[3], Nadine Sen Nkwe[1], Daméhan Tchelougou [1], Maxime Uriarte[1], Eric Bonneil [4], Derek Ceccarelli [2], Nazar Mashtalir[1], Mika Tanji[5], Jean-Yves Masson[6], Pierre Thibault[4], Frank Sicheri[2], Haining Yang[5], Michele Carbone[5], Marc Therrien[3,7] & El Bachir Affar[1]

The tumor suppressor and deubiquitinase (DUB) BAP1 and its *Drosophila* ortholog Calypso assemble DUB complexes with the transcription regulators Additional sex combs-like (ASXL1, ASXL2, ASXL3) and Asx respectively. ASXLs and Asx use their DEUBiquitinase ADaptor (DEUBAD) domain to stimulate BAP1/Calypso DUB activity. Here we report that monoubiquitination of the DEUBAD is a general feature of ASXLs and Asx. BAP1 promotes DEUBAD monoubiquitination resulting in an increased stability of ASXL2, which in turn stimulates BAP1 DUB activity. ASXL2 monoubiquitination is directly catalyzed by UBE2E family of Ubiquitin-conjugating enzymes and regulates mammalian cell proliferation. Remarkably, Calypso also regulates Asx monoubiquitination and transgenic flies expressing monoubiquitination-defective Asx mutant exhibit developmental defects. Finally, the protein levels of ASXL2, BAP1 and UBE2E enzymes are highly correlated in mesothelioma tumors suggesting the importance of this signaling axis for tumor suppression. We propose that monoubiquitination orchestrates a molecular symbiosis relationship between ASXLs and BAP1.

[1] Maisonneuve-Rosemont Hospital Research Center and Department of Medicine, University of Montréal, Montréal, QC H3C 3J7, Canada. [2] Lunenfeld-Tanenbaum Research Institute, Sinai Health System, Toronto, ON M5G 1X5, Canada. [3] Institute for Research in Immunology and Cancer, Laboratory of Intracellular Signaling, University of Montréal, Montréal, QC H3T 1J4, Canada. [4] Institute for Research in Immunology and Cancer, Laboratory of Proteomics and Bioanalytical Mass Spectrometry, University of Montréal, Montréal, QC H3T 1J4, Canada. [5] University of Hawaii Cancer Center, University of Hawaii, Honolulu, HI 96813, USA. [6] CHU de Quebec Research Center (Oncology Axis), Laval University Cancer Research Center, 9 McMahon, Quebec, PQ G1R 2J6, Canada. [7] Département de pathologie et biologie cellulaire, University of Montréal, Montréal, QC H3C 3J7, Canada. These authors contributed equally: Salima Daou, Haithem Barbour. Correspondence and requests for materials should be addressed to M.T. (email: marc.therrien@umontreal.ca) or to E.B.A. (email: el.bachir.affar@umontreal.ca)

The Ubiquitin (Ub) C-terminal hydrolase (UCH) family of deubiquitinases (DUBs) contains four members including UCH37 (UCHL-5) and BAP1 which share high similarity in the catalytic domain (UCH) and the C-terminal region, termed the UCH37-like domain (ULD) or the C-terminal domain (CTD), in UCH37 and BAP1 respectively[1]. BAP1 is a tumor suppressor inactivated in numerous cancers[2,3], and ablation of this gene in mice also results in tumor development[4–7]. BAP1 is localized predominantly in the nucleus where it assembles large multi-protein complexes containing transcription factors and co-factors including Host Cell Factor 1 (HCF-1), O-linked N-acetyl-Glucosamine Transferase (OGT), Additional Sex Comb Like proteins ASXL1 and ASXL2 (ASXL1/2), as well as transcription factors[8–10]. BAP1 regulates gene expression[10], double-strand DNA repair[11,12] and DNA replication[13]. Due to its multiple functions, BAP1 is tightly regulated. Indeed, the subcellular localization of BAP1 is coordinated by the E3 ligase UBE2O which promotes BAP1 cytoplasmic retention[14]. In the cytoplasm, BAP1 interacts with type 3 inositol-1,4,5-trisphosphate receptor (IP3R3) at the endoplasmic reticulum and promotes Ca2+ signaling and apoptosis in response to stress[15].

Calypso, the Drosophila ortholog of mammalian BAP1, associates with the transcription regulator Additional Sex Comb (Asx) forming a Polycomb Group (PcG) complex (PR-DUB complex) which stimulates the DUB activity of Calypso and represses PcG target genes including the Hox gene Ultrabithorax (Ubx)[16]. Although PR-DUB deubiquitinates monoubiquitinated histone H2A on K118 (H2AK119 in vertebrates, hereafter H2Aub), the functional significance of this deubiquitination event remains unclear, as mutation of K118 of H2A does not impact the repression of Ubx[17]. Interestingly, Asx is an atypical PcG factor, since it is required to maintain transcriptional silencing as well as activation[18,19]. Mammalian ASXLs (ASXL1, ASXL2, and ASXL3) are highly conserved paralogs that diverged from Asx but preserved their interaction with BAP1[8–10]. ASXL1 and ASXL2 are also reported to function with both co-repressors and co-activators[20–22], and are recurrently mutated in cancer[23–25]. ASXLs and Asx contain an uncharacterized N-terminal domain (ASXN), the DEUBiquitinase Adaptor (DEUBAD, also termed ASXM) domain, as well as a C-terminal Plant Homeo Domain (PHD) finger (Fig. 1a). The DEUBAD domain is also found in UCH37-interacting partners, RPN13 (ADRM1) and INO80G (NFRKB), components of the proteasome regulatory particle and the INO80 chromatin remodeling complex respectively[26]. Interestingly, while UCH37 is inhibited by INO80G, this DUB is activated by RPN13[27–30]. Structural studies showed that the DEUBAD of RPN13 interacts with the ULD and this ensures coordination of UCH domain binding with ubiquitin, including the repositioning of the cross-over loop, thus promoting catalysis. In contrast, the DEUBAD of INO80G uses similar structural determinants to engage a conformational configuration incompatible with catalysis, thus inhibiting UCH37[31,32].

ASXLs use their respective DEUBAD domain to assemble distinct and mutually exclusive complexes with the CTD of BAP1[33,34]. This interaction stimulates ubiquitin binding and DUB activity[16,33,34] (Fig. 1b). Interestingly, deletion of the DEUBAD domain results in the disruption of BAP1/ASXLs interaction and leads to decreased ASXL2, but not ASXL1, protein levels[33]. This suggests that interaction through the DEUBAD domain promotes ASXL2 stability. In contrast, ASXL1 was shown to be ubiquitinated on its DEUBAD domain and this event promotes its degradation[35]. Nonetheless, it remains unclear how ASXL1 and ASXL2 are regulated by ubiquitination and whether BAP1 is involved in these events. Our study shows that monoubiquitination of DEUBAD by UBE2E enzymes, is a highly conserved regulatory mechanism of BAP1 DUB activity.

Monoubiquitination of the DEUBAD stabilizes ASXL2, stimulates BAP1 DUB activity and regulates mammalian cell proliferation. Moreover, expression of a monoubiquitination-defective Asx mutant alters Drosophila development. Thus, BAP1 promotes the ubiquitination of its co-factor ASXL2 regulating its stability, and this in turn regulates the enzymatic activity and function of the DUB complex.

## Results

**BAP1 promotes ASXL2 monoubiquitination on its DEUBAD domain.** Because the interaction between BAP1 and ASXL2 is important for their stability and tumor suppression[33], we sought to investigate the potential role of ubiquitination in coordinating BAP1/ASXL2 stability and function. First, we co-transfected Myc-ASXL2 in HEK293T cells with HA-Ub and Flag-BAP1, and immunoprecipitated (IP) ASXL2 protein to detect its ubiquitination state. Immunoblotting with anti-HA (Ub) revealed a distinct ASXL2 protein band suggesting its monoubiquitination (Fig. 1c). This signal was strongly enhanced upon co-expression of ASXL2 with BAP1. As ASXL2 is around 200 kDa, we used GFP-Ub to ensure a distinct molecular weight shift and validated that overexpressed ASXL2 is monoubiquitinated in BAP1-dependent manner (Fig. 1d). Of note, GFP-Ub was not conjugated as efficiently as Ub fused to a small tag (e.g., HA or Myc). This could be caused by the bulky size of the GFP tag which might sterically interfere with ubiquitin charging/ligation. In addition, depletion of BAP1 by shRNA in U-2 OS cells resulted in reduced levels of endogenous ASXL2 with a noticeable band shift suggesting that the majority of ASXL2 is constitutively mono-ubiquitinated (Fig. 1e). The decrease of ASXL2 protein levels might suggest its polyubiquitination and proteasomal degradation as a consequence of the loss of interaction with BAP1. Note that the very faint band occasionally observed below the major band of ASXL2 corresponds to its unmodified form (Fig. 1f). Depletion of ASXL2 using siRNA validated the identity of the two detected bands of this factor (Fig. 1f).

To identify the regions/domains of ASXL2 required for its monoubiquitination, we used ASXL2 deletion mutants lacking known domains (ASXL2ΔASXN, ASXL2ΔDEUBAD, and ASXL2ΔPHD) that we overexpressed with GFP-Ub and Flag-BAP1 and found that the DEUBAD domain is necessary for ASXL2 monoubiquitination (Fig. 1g). Strikingly, we co-expressed Myc-DEUBAD (~21 kDa) with Flag-BAP1 and HA-Ub and revealed that the DEUBAD domain alone is monoubiquitinated in BAP1-dependent manner (Fig. 1h). We then conducted a large-scale expression of Flag-ASXL2 in HEK293T and purified this factor for mass spectrometry (MS) analysis. We identified several ubiquitination sites on the DEUBAD domain (Supplementary Fig. 1a, b). Of note, several ubiquitination sites have been previously identified in systematic proteomics studies. Notably, the K370 ubiquitination site we identified (Fig. 1i), has been repeatedly observed (Supplementary Fig. 1b). To validate the MS data, we generated lysine to arginine mutants of Myc-DEUBAD, and identified the K370 residue as the key BAP1-dependent monoubiquitination site (Fig. 1j). Note that an upper band shift of BAP1 is observed following expression of Myc-DEUBAD K370R suggesting its potential modification. This further highlights the possible intricate relationship between DEUBAD monoubiquitination and BAP1.

**DEUBAD monoubiquitination is highly conserved.** We transfected Myc-ASXL1 in HEK293T cells with and without HA-Ub and Flag-BAP1 and found that this factor is also ubiquitinated in a BAP1-dependent manner (Supplementary Fig. 1c). MS analysis revealed a major ubiquitination site within ASXL1 DEUBAD and

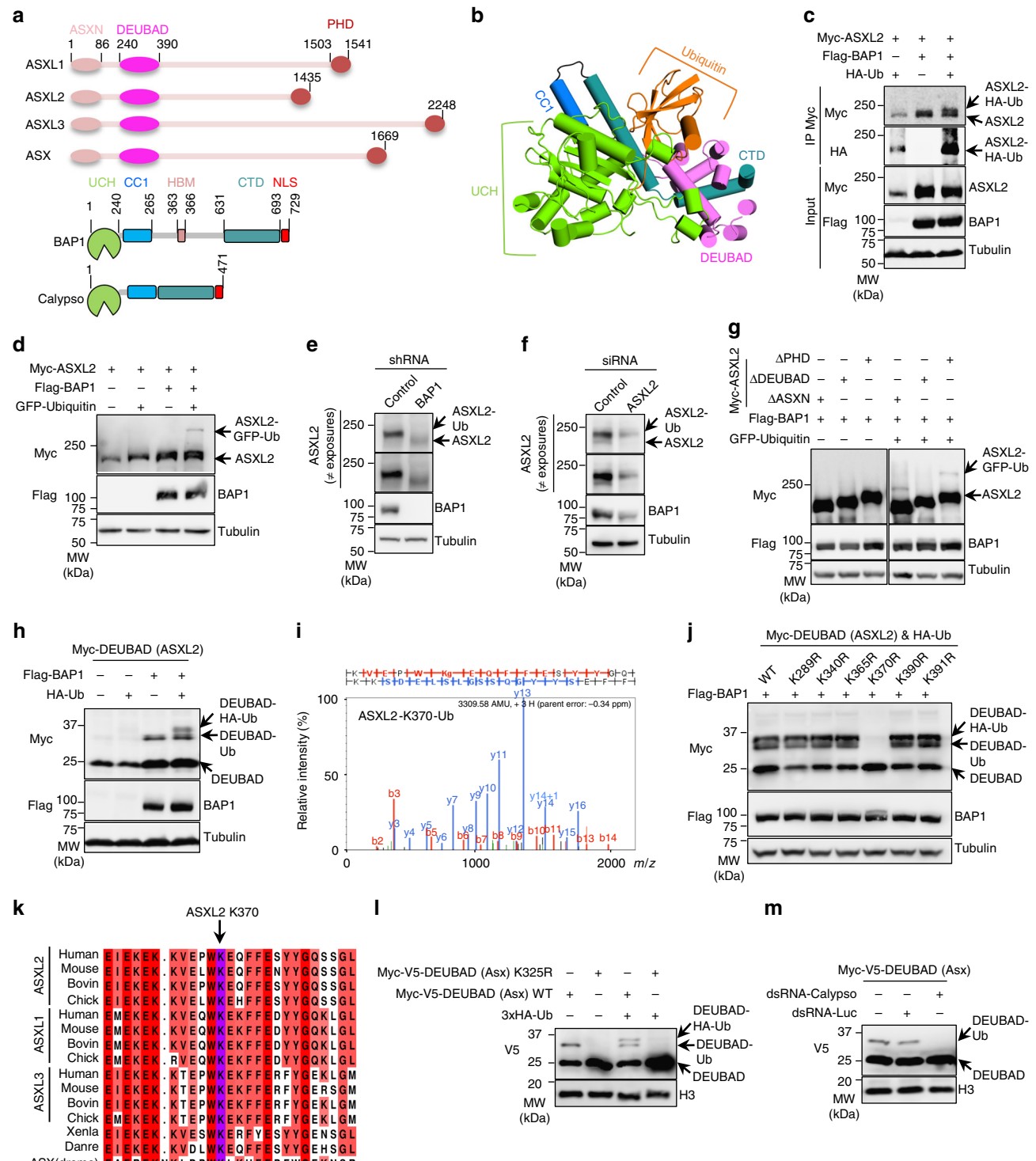

further validated K351 as the main site of monoubiquitination (Supplementary Fig. 1b, d, e). Next, we aligned vertebrate ASXLs DEUBADs with that of Asx and found that the monoubiquitination site is conserved throughout evolution (Fig. 1k). We noted that ASXL3 was also found in proteomics studies to be ubiquitinated on K350 which corresponds to the highly conserved monoubiquitination site of ASXL2 DEUBAD (Supplementary Fig. 1b and Fig. 1k). Thus, we transfected Myc-DEUBAD and validated this domain is indeed monoubiquitinated on K350 in BAP1-dependent manner (Supplementary Fig. 1f). These results prompted us to use *Drosophila* S2 cells to express Myc-V5-tagged

DEUBAD of Asx and determine its ubiquitination state. Expression of the DEUBAD alone induced a mobility shift suggesting its monoubiquitination (Fig. 1l). The upper band was abolished following expression of Myc-V5-DEUBAD K325R mutant (K370 in human ASXL2). Co-transfection of Myc-V5-DEUBAD or K325R mutant with HA-Ub further validated the monoubiquitination of Asx DEUBAD. Importantly, dsRNA depletion of endogenous Calypso strongly reduced DEUBAD monoubiquitination (Fig. 1m). Moreover, overexpression of Calypso and full length Asx induced a band shift of the latter, which is typical of monoubiquitination (Supplementary Fig. 2a).

**Fig. 1** ASXL2 is monoubiquitinated on DEUBAD in a BAP1-dependent manner. **a** Schema representation of ASXLs protein family, Asx and BAP1/Calypso proteins. **b** Cartoon representation of BAP1/Ub/DEUBAD (DEUBAD of ASXL2) homology structure, based on the UCH37/Ub/DEUBAD (DEUBAD of RPN13) crystal structure (PDB, 4UEL). **c** BAP1 enhances ubiquitination of ASXL2. HEK293T cells were transfected with Myc-ASXL2, Flag-BAP1 or HA-Ub vectors and subjected to immunoprecipitation and immunoblotting. $n = 3$ biological replicates. **d** ASXL2 is monoubiquitinated in BAP1-dependent manner. HEK293T cells were transfected as indicated and ASXL2 ubiquitination with GFP-Ub was analyzed by immunoblotting. $n = 4$ biological replicates. **e** Decrease of ASXL2 protein levels in U-2 OS cells stably expressing shRNA of *BAP1*. $n = 5$ biological replicates. **f** Depletion of ASXL2 using siRNA in U-2 OS cells. $n = 5$ biological replicates. **g** DEUBAD is required for ASXL2 monoubiquitination. HEK293T cells were co-transfected with Flag-BAP1 and the corresponding deletion mutants constructs of Myc-ASXL2 in presence or not of GFP-Ub and subjected to immunoblotting. $n = 4$ biological replicates. **h** The DEUBAD is sufficient for its monoubiquitination in BAP1-dependent manner. The indicated constructs were transfected in HEK293T cells and DEUBAD ubiquitination was analyzed. $n = 4$ biological replicates. **i** Mass spectrometry (MS) spectrum indicating Ub remnant on Lysine 370 of ASXL2. **j** Lysine 370 is the BAP1-dependent monoubiquitination site of ASXL2. HEK293T cells were transfected with the corresponding lysine mutants of DEUBAD (ASXL2) and used for immunoblotting. $n = 3$ biological replicates. **k** Conservation of the ASXLs ubiquitination site. Sequence alignment of ASXLs orthologs (ASXL2 K370 site highlighted in purple). **l** The DEUBAD of Asx is monoubiquitinated in *Drosophila*. S2 cells were transfected with Myc-V5-DEUBAD (Asx) and Myc-V5-DEUBAD (Asx) K325R expression vectors and subjected to immunoblotting. $n = 3$ biological replicates. **m** *Drosophila* DEUBAD K325 is monoubiquitinated in Calypso-dependent-manner. Myc-V5-DEUBAD (Asx) was co-transfected with control or Calypso dsRNA in S2 cells and its monoubiquitination levels were determined by immunoblotting. $n = 3$ biological replicates. Tubulin was used as a loading control for panels **c–h** and **j**. Histone H3 was used as a loading control for panels **l**, **m**

Calypso also promoted the monoubiquitination of the wildtype (WT) DEUBAD, but not its corresponding K325R mutant (Supplementary Fig. 2b). Interestingly, we also found that over-expression of BAP1 or Calypso in HEK293T cells promoted the monoubiquitination of the DEUBAD of Asx and ASXL2 on K325 and K370, respectively (Supplementary Fig. 2c, d). Finally, we sought to test whether a similar mechanism of DEUBAD monoubiquitination, involving UCH37, could be observed for INO80G and RPN13. We did not identify the potential ubiquitination site in RPN13 or INO80G DEUBADs through protein sequence alignment with ASXL2 DEUBAD (Supplementary Fig. 2e). We also expressed RPN13 or INO80G DEUBADs with UCH37 WT or its corresponding catalytic dead mutant, UCH37 C88A in HEK293T cells, and analyzed potential DEUBAD modifications. Although we observed additional upper bands for RPN13 and INO80G that can suggest DEUBAD ubiquitination, these patterns did not change upon expression of UCH37 (Supplementary Fig. 2f). Thus, we concluded that the DUB-dependent monoubiquitination of DEUBAD is specific to Asx/ASXLs and is highly conserved from *Drosophila* to human.

**Proper BAP1 assembly promotes DEUBAD monoubiquitination**. Intramolecular interactions between the UCH and the CTD of BAP1, as well as interaction of the CTD with the DEUBAD of ASXLs are prerequisite for the assembly of a DUB competent complex[14,33,34] (Fig. 1a, b). In particular, a coiled-coil 1 domain (CC1) juxtaposed to the UCH establishes an interaction with a coiled-coil domain (CC2) within the CTD to recruit the DEUBAD in close proximity to the catalytic domain. The middle region between CC1 and CTD containing the HCF-1 binding motif (HBM) is dispensable for DUB catalysis[33,34]. To define whether these interactions are necessary for DEUBAD monoubiquitination, we used several Flag-BAP1 deletion mutants targeting the UCH, CC1 and CTD that we expressed with the DEUBAD of ASXL2 (Fig. 2a). Deletion of CC1 or CTD resulted in a strong decrease of DEUBAD monoubiquitination (Fig. 2b). Importantly, a cancer-associated mutation of BAP1 in its CTD, BAP1$^{\Delta R666-H669}$, that abrogates its interaction with ASXL1 and ASXL2[33], also failed in promoting DEUBAD monoubiquitination (Fig. 2a, b). Further demonstrating the functional inter-dependency of BAP1 domains, we found that co-expression of GFP-UCH-CC1, Myc-CTD, and Myc-DEUBAD were minimally required to promote DEUBAD monoubiquitination (Fig. 2c). The basal levels of DEUBAD monoubiquitination occasionally observed is likely due to endogenous BAP1. Of note, expression of

the UCH domain often results in an additional upper band suggesting its modification (Fig. 2c). Finally, monoubiquitination of the DEUBAD is not impaired when this domain is expressed with BAP1$^{\Delta HBM}$ suggesting that BAP1 interaction with HCF-1 is not required for this modification (Fig. 2d). Thus, our results show that the cellular levels of monoubiquitinated DEUBAD are strongly increased or maintained through the assembly with BAP1 of a DUB competent complex.

**DEUBAD monoubiquitination does not require BAP1 DUB activity**. Biochemical and structural data indicated that BAP1 interaction with the DEUBAD generates a Composite Ub Binding Interface (CUBI) that promotes DUB activity[33,34]. We found that monoubiquitination of ASXL2 DEUBAD was completely abrogated following deletion of the UCH (Fig. 2d). However, mono-ubiquitination of DEUBAD was induced to a similar extent by WT BAP1 or a catalytically inactive (C91S) mutant (Fig. 2d). Thus, we hypothesized that Ub binding by the BAP1/DEUBAD complex without actual catalysis is important for DEUBAD monoubiquitination and we sought to investigate this possibility. Based on the structure of UCHL-5 in complex with Ub[31,32], we generated several point mutants within the Ub-binding interface of BAP1 and tested their impact on DEUBAD monoubiquitination. As expected, these mutants are either not modified or very poorly modified by HA-Ub-Vinyl Methyl Ester (HA-Ub-VME)[36,37], a Ub activity probe that covalently binds the catalytic cysteine of DUBs (Fig. 2e). However, these mutants were as effective as the wild type BAP1 in promoting DEUBAD mono-ubiquitination (Fig. 2f). These results indicate that mono-ubiquitination of the DEUBAD does not involve the catalytic activity of BAP1 or its ability to bind Ub, yet involves the assembly of a DUB competent complex.

**Opposing roles of ASXL2 K370 ubiquitination**. ASXL2 protein levels and its monoubiquitination state are dependent on BAP1 (Fig. 1c–e). Thus, we evaluated how monoubiquitination regulates ASXL2 protein stability. Immunoblotting of HEK293T or U-2 OS cell extracts showed one major upper band of endogenous ASXL2 (monoubiquitinated form) and one very faint band below (non-ubiquitinated form), both of which increased in intensity following proteasome inhibition with MG132 (Fig. 3a, arrows). Treatment of HEK293T cells with cycloheximide (CHX) revealed that the levels of monoubiquitinated ASXL2 are hardly reduced up to 8 h post-treatment, while the non-ubiquitinated form disappeared within 1 h of treatment (Fig. 3b, arrows). These

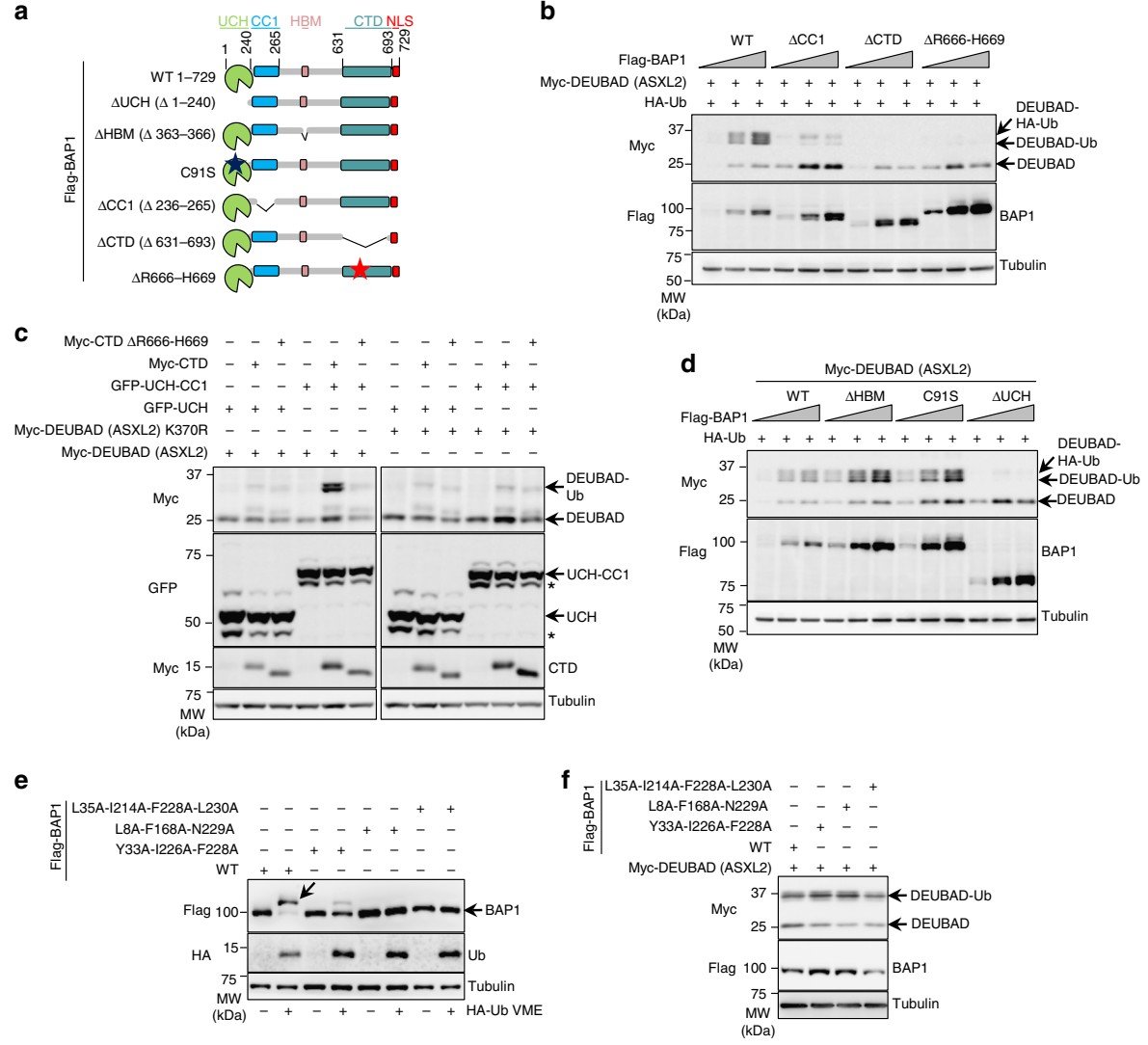

**Fig. 2** Monoubiquitination of DEUBAD requires BAP1 intramolecular interactions. **a** Representation of the different BAP1 mutant forms used for experiments done in the panels **b** and **d**. **b** Disruption of the intra-molecular interactions between the catalytic (UCH) and non-catalytic (CC1 and CTD) domains of BAP1 as well as inhibition of its interaction with DEUBAD by the R666-H669 cancer associated mutation impair DEUBAD domain monoubiquitination. HEK293T cells were transfected with BAP1 and its mutant forms as indicated in presence of Myc-DEUBAD (ASXL2) and subjected to western blotting. $n = 4$ biological replicates. **c** Reconstitution of the intramolecular interactions of BAP1 promotes DEUBAD monoubiquitination. Co-transfection of Myc-DEUBAD (ASXL2) or Myc-DEUBAD (ASXL2) K370R with the corresponding GFP BAP1- or Myc-BAP1- fragments fusion constructs in HEK293T and cell extracts were used for western blotting. The star indicates a possible degradation product. The band upper UCH or UCH-CC1 represents a potential post-translational modification of this domain. $n = 2$ biological replicates. **d** Monoubiquitination of the DEUBAD domain following expression of BAP1$^{\Delta HBM}$, BAP1$^{\Delta UCH}$, or BAP1 catalytic dead (C91S) constructs in HEK293T cells. Increased amounts of the different Flag-BAP1 constructs were used in presence of HA-Ub and Myc-DEUBAD (ASXL2), then DEUBAD monoubiquitination was visualized by western blotting. $n = 4$ biological replicates. **e** Validation of Ub binding mutants of BAP1. Flag-BAP1 and its different mutant forms were transfected in HEK293T cells. Cell lysates were labeled with HA-Ub-VME DUB probe and analyzed by immunoblotting. $n = 2$ biological replicates. **f** BAP1 ubiquitin binding is not required for DEUBAD monoubiquitination. HEK293T cells were transfected with BAP1 Ub binding mutants in presence of Myc-DEUBAD (ASXL2) and harvested for immunoblotting. $n = 2$ biological replicates. Tubulin was used as a loading control for panels **b**–**f**

results suggest that monoubiquitinated ASXL2 is relatively stable, while a fast turnover impedes robust detection of its non-ubiquitinated form. The DNA replication factor, Cdc6, was included as an internal control for inhibition of proteasomal degradation or protein synthesis[38]. To enhance the detection of unmodified ASXL2, we overexpressed this factor by viral trans-duction in U-2 OS cells and found that both non-ubiquitinated and monoubiquitinated forms of HA-ASXL2 were readily detected (Fig. 3c, d and Supplementary Fig. 3a, arrows). Because of overexpression, the protein levels of the unmodified ASXL2 are predominant relative to the monoubiquitinated form. As

expected, overexpression of HA-ASXL2 K370R produced only one major band (Fig. 3c, d). Interestingly, the protein levels of WT HA-ASXL2 were, in general, lower than the corresponding K370R mutant (Fig. 3c, d), and proteasome inhibition with MG132 stabilized ASXL2 up to the levels of the K370R mutant (Fig. 3c). Moreover, following CHX chase, and similarly to endogenous ASXL2, the levels of the monoubiquitinated form of ASXL2 hardly decreased within 4 h, while the non-ubiquitinated form almost completely disappeared within 1 h of treatment (Fig. 3d and Supplementary Fig. 3a). As expected, the protein levels of ASXL2 K370R mutant showed a less pronounced

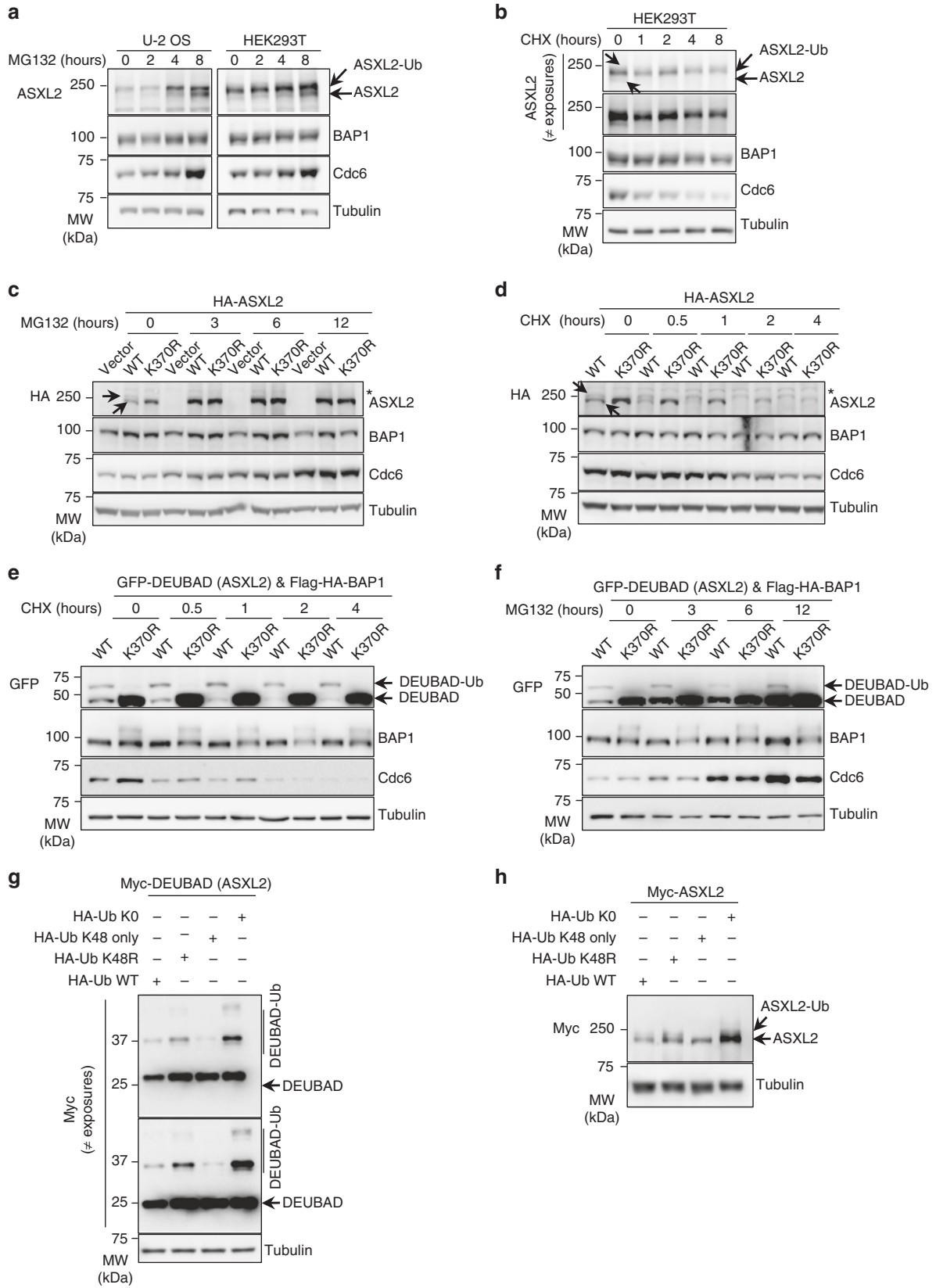

decrease than the non-ubiquitinated form (Fig. 3d). Of note, no major changes were observed on the levels of BAP1 following MG132 or CHX treatments. To further substantiate these results, we generated U-2 OS cells stably expressing Flag-HA-BAP1 with either GFP-DEUBAD of ASXL2 or the corresponding K370R

mutant. BAP1 overexpression results in elevated levels of DEU-BAD, allowing an improved detection of both monoubiquitinated and unmodified forms of the DEUBAD. Consistent with the data obtained with ASXL2, we found that protein levels of GFP-DEUBAD K370R mutant were significantly higher than the WT

**Fig. 3** Dual role of K370 ubiquitination on ASXL2 protein stability. **a** ASXL2 protein levels are accumulated following proteasome inhibition. U-2 OS or HEK293T cells were treated with 20 μM of MG132 for the indicated times and harvested for immunoblotting. $n = 2$ biological replicates. **b** HEK293T cells were treated with 20 μg/ml of cycloheximide (CHX) as indicated and ASXL2 levels were determined by immunoblotting. The arrows in panels **a** and **b** indicate the unmodified and the monoubiquitination forms of ASXL2. $n = 2$ biological replicates. **c** The non-modified form of ASXL2 is rapidly accumulated relative to its monoubiquitinated form following proteasome inhibition. U-2 OS cells were infected with lentiviral expressing vectors for ASXL2 or ASXL2 K370R then treated with 20 μM of MG132 for the indicated times and used for immunoblotting. $n = 2$ biological replicates. **d** Monoubiquitination of ASXL2 and mutation of K370 maintains its stability. U-2 OS cells stably expressing HA-ASXL2 or HA-ASXL2 K370R were treated with CHX as indicated and analyzed by immunoblotting as in **c**. $n = 2$ biological replicates. The arrows in panels **c** and **d** indicate the unmodified and the monoubiquitination forms of HA-ASXL2. **e** Monoubiquitination of DEUBAD and mutation of its K370 maintain its stability. U-2 OS cells stably expressing GFP-DEUBAD (ASXL2) or GFP-DEUBAD (ASXL2) K370R were treated with CHX as indicated. Cell lysates were used for immunoblotting. $n = 3$ biological replicates. **f** Unmodified DEUBAD is strongly accumulated relative to the monoubiquitinated form, following proteasome inhibition. U-2 OS cells stably expressing GFP-DEUBAD (ASXL2) or GFP-DEUBAD (ASXL2) K370R were treated with 20 μM of MG132 for the indicated times. Cell extracts were analyzed by immunoblotting. $n = 3$ biological replicates. **g** Further evidence that DEUBAD (ASXL2) K370 is also a site for Ub chain extension and degradation. HEK293T cells were transfected with Myc-DEUBAD (ASXL2) and HA-tagged Ub vectors and harvested for immunoblotting. $n = 2$ biological replicates. **h** ASXL2 stability is regulated by polyubiquitination inducing its degradation. HEK293T cells were transfected with Myc-ASXL2 and HA-tagged Ub vectors as indicated and subjected to immunoblotting. $n = 2$ biological replicates. Tubulin was used as a loading control for all panels

form (Fig. 3e, f). Note the upper band of BAP1, suggestive of its modification, observed following expression of DEUBAD K370R. Following treatment with CHX, the unmodified form of GFP-DEUBAD was very unstable comparatively to the mono-ubiquitinated or the K370R mutant forms (Fig. 3e). We also observed that MG132 treatment leads to a strong increase in the protein levels of unmodified GFP-DEUBAD which become comparable to the levels of the corresponding K370R mutant (Fig. 3f).

Based on our results altogether, we anticipated that mono-ubiquitination of ASXL2 on K370 stabilizes this factor and that Ub chain extension might occur on the same site to promote its degradation. To further support this, we co-transfected Myc-DEUBAD of ASXL2 with several HA-Ub constructs including the K0 mutant (all 7K to R). We also included K48R and K48 only mutants. We observed that the monoubiquitination signal of the DEUBAD domain was increased when this domain is co-expressed with Ub mutants that cannot assemble degradation-inducing Ub chains (HA-Ub K48R and HA-Ub K0) (Fig. 3g). Moreover, an additional upper band of DEUBAD becomes readily noticeable when this domain is expressed with these Ub chain elongation-defective mutants. The upper band likely corresponds to HA-Ub K48R or HA-Ub K0 conjugated to endogenous Ub already ligated to DEUBAD. We also observed that protein levels of full length ASXL2 were higher following co-expression of Myc-ASXL2 with K48R and K0 mutants (Fig. 3h). Finally, we noticed that the levels of the monoubiquitinated forms of DEUBAD or ASXL2 were reduced when HA-Ub K48 only mutant is overexpressed with Myc-DEUBAD or Myc-ASXL2 (Fig. 3g, h).

**Validation of Ub chain extension on ASXL2 K370**. We reasoned that ASXL2 stability might be regulated by DUBs whose inhibition might result in polyubiquitin chain extension on K370. First, we assembled a library of siRNA targeting human DUBs (Supplementary Table 1), and conducted a loss-of-function screen in U-2 OS cells stably expressing Flag-HA-BAP1 and GFP-DEUBAD of ASXL2. We identified several DUB candidates whose depletion resulted in decreased or increased DEUBAD protein levels relative to the non-target (NT) control siRNA (Fig. 4a, b and Supplementary Fig. 3b). BAP1 protein levels were less affected than those of the DEUBAD domain. Interestingly, the non-ubiquitinated form of DEUBAD showed more variation than the monoubiquitinated form (Supplementary Fig. 3b), and this correlated with their respective degree of stability. The protein levels of YY1, used as a control, were hardly affected by DUB depletion. Notably, we also observed that depletion of the

proteasome associated DUB PSMD14 or its co-factor PSMD7, results in a strong increase in the protein levels of the unmodified form of DEUBAD (Fig. 4a, b and Supplementary Fig. 3b). In addition, we observed the appearance of multiple upper bands suggesting DEUBAD polyubiquitination (Supplementary Fig. 3b). Using a different set of PSMD14 siRNAs, we observed a strong accumulation of the unmodified DEUBAD and the appearance of additional upper bands (Fig. 4c). Moreover, the upper bands of DEUBAD were completely abrogated for the K370R mutant, thus further demonstrating that this monoubiquitination site can be subjected to polyubiquitination (Fig. 4c). We concluded that monoubiquitination and polyubiquitination are two distinct events that target K370 of ASXL2 DEUBAD.

**UBE2E enzymes catalyze K370 monoubiquitination**. As BAP1 promotes ASXL2 monoubiquitination in vivo, we evaluated whether BAP1 with E2-conjugating enzymes are sufficient to direct this event in vitro. We used an in vitro assay system whereby recombinant His-BAP1/MBP-DEUBAD (of ASXL2) complex was purified from bacteria and incubated with a panel of 32 recombinant UBE2s in the presence of the UBA1 E1 and Ub/ATP. Strikingly, we found that only UBE2E1, UBE2E2, and UBE2E3 monoubiquitinate the DEUBAD domain (Fig. 5a and Supplementary Fig. 4b). UBE2E family is highly conserved and is characterized by a distinct N-terminal extension which is highly variable between paralogs (Supplementary Fig. 4a). Interestingly, UBE2E family enzymes were shown to catalyze direct ubiquitination of SETDB1 without additional E3 ligases[39].

Our results, obtained from cell-based studies, suggest that BAP1 might either directly stimulate ASXL2 monoubiquitination or preserve monoubiquitinated ASXL2 from degradation. In further dissecting the mechanism of catalysis in vitro, we found that UBE2E3 monoubiquitinates recombinant DEUBAD either alone or in complex with BAP1, and mutation of K370 abolishes DEUBAD monoubiquitination (Fig. 5b). This result suggests that, in vivo, UBE2Es might monoubiquitinate free ASXL2 as well as ASXL2 in complex with BAP1. Therefore, BAP1 interaction would protect monoubiquitinated ASXL2 form Ub chain extension resulting in a predominantly higher levels of its modified form.

On the other hand, we found that BAP1 enzymatic activity and its ability to bind Ub do not impact DEUBAD monoubiquitination in vivo (Fig. 2e, f). However, other cellular components might interfere with this event. Hence, we used recombinant protein products of several BAP1 mutants in an in vitro reaction with defined components, and demonstrated that its enzymatic activity and ability to bind Ub do not impact UBE2E3-mediated

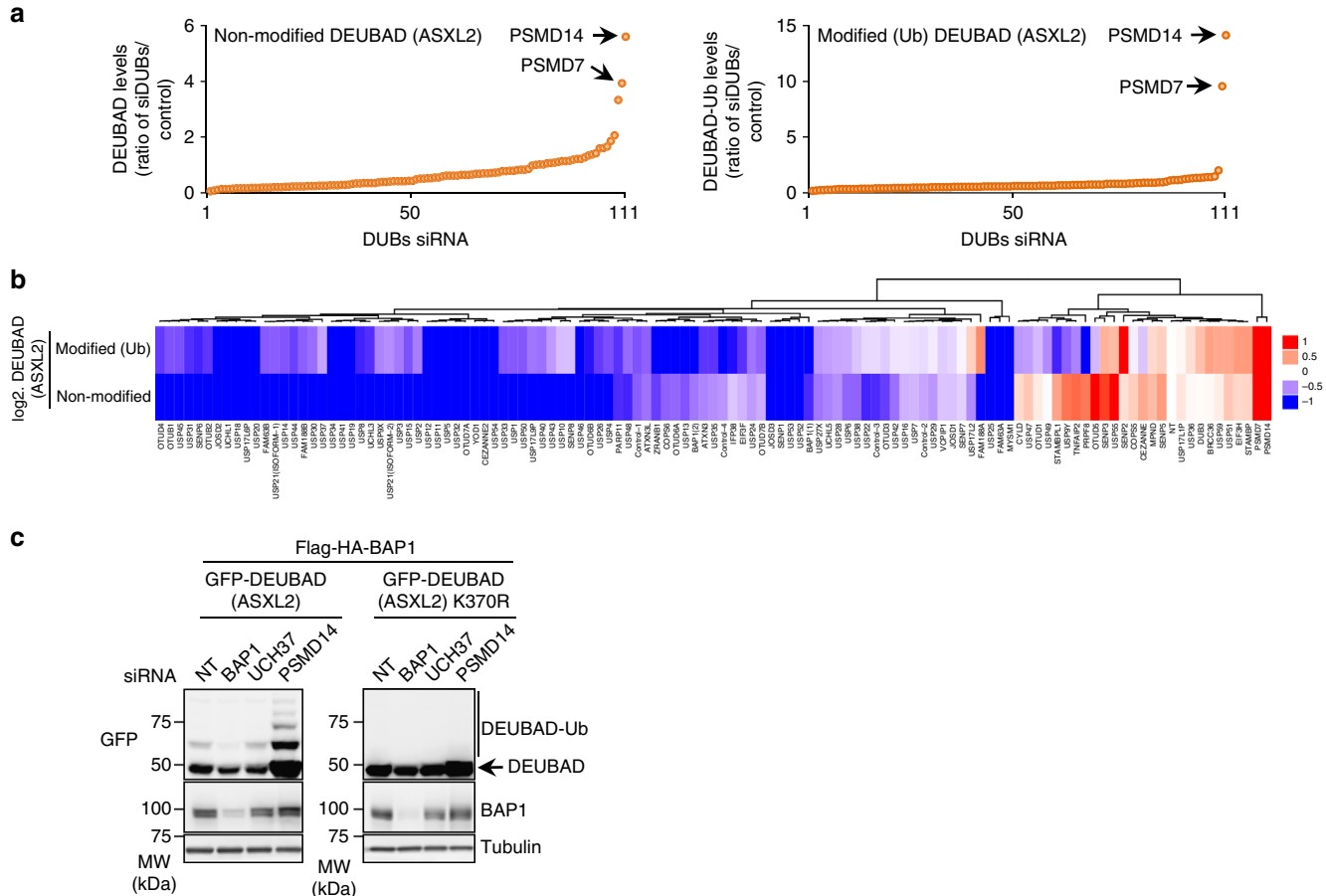

**Fig. 4** K370 is the site of mono- and polyubiquitination. **a** DEUBAD protein levels are affected by DUB depletion. U-2 OS cells stably expressing Flag-HA-BAP1 and GFP-DEUBAD (ASXL2) were transfected with siRNA library for all human DUBs (see also Supplementary Fig. 3b). Densitometry analysis of protein bands in DUB siRNA versus non-target (NT) siRNA control were conducted. The arrows indicate the top DUBs hits that increased the corresponding protein levels. $n = 1$ biological sample. **b** Comparison of changes in band intensity between unmodified and monoubiquitinated forms of DEUBAD (ASXL2). **c** Validation that K370 is the site of mono- and poly-ubiquitination of DEUBAD (ASXL2). U-2 OS cells stably expressing Flag-HA-BAP1 and either GFP-DEUBAD (ASXL2) or GFP-DEUBAD (ASXL2) K370R were transfected with siRNA NT control or siRNAs targeting different DUBs and harvested for western blotting as indicated. $n = 2$ biological replicates. Tubulin was used as a loading control for panel **c**

DEUBAD monoubiquitination (Fig. 5c, d), thus corroborating our data obtained in vivo.

To further examine the roles of UBE2E family and other E2 Ub-conjugating enzymes in regulating ASXL2 stability, we assembled a siRNA library of human E2 enzymes (Supplementary Table 2), and used U-2 OS cells stably expressing GFP-DEUBAD of ASXL2 to conduct a loss-of-function screen followed by immunoblotting. We expressed GFP-DEUBAD only as we found that UBE2E3 could directly monoubiquitinate this domain in the absence of BAP1 (Fig. 5b). Depletion of several E2s resulted in a decrease of DEUBAD protein levels relative to the non-target control siRNA (Supplementary Fig. 5a, b). Strikingly, depletion of UBE2E3 resulted in the most significant increase of the protein levels of the unmodified DEUBAD, while a slight effect was observed on the monoubiquitinated form (Supplementary Fig. 5a, b). This result is consistent with our earlier observations and suggests a model whereby UBE2E3 might monoubiquitinate two pools of DEUBAD: i) the free DEUBAD, which would become primed for degradation, and ii) the DEUBAD in complex with BAP1, which would become protected from degradation. Finally, we note that UBE2O, which we previously showed to directly monoubiquitinate BAP1 nuclear localization site (NLS)[14], is not involved in DEUBAD monoubiquitination (Supplementary Fig. 5c, d). Altogether, these results suggest that UBE2E family

enzymes are responsible for DEUBAD monoubiquitination and appear to dictate, depending on its interaction with BAP1, different outcomes on ASXL2.

**UBE2Es promote ASXL2 degradation or stabilization.** To further examine the role of UBE2E family in regulating ASXL2, we first transfected U-2 OS cells stably expressing GFP-DEUBAD alone with additional siRNAs oligos for individual UBE2Es and conducted CHX chase. We confirmed that UBE2E3 and UBE2E2 knockdown resulted in a substantial increase of DEUBAD protein levels above that of the non-target siRNA control (Fig. 6a and Supplementary Fig. 5e). Note that siRNA of UBE2E2 also partially depleted UBE2E3 (Supplementary Fig. 5e). Combined depletion of all three UBE2Es resulted in an increase of DEUBAD protein levels comparable to that observed following UBE2E2 or UBE2E3 knockdown (Fig. 6b). Moreover, DEUBAD proteolysis following CHX chase was strongly reduced when all three enzymes were depleted. Also, combined knockdown of UBE2Es resulted in increased GFP-DEUBAD protein to a similar level to that of the DEUBAD K370R (Fig. 6b). Next, we expressed Flag-UBE2E-1, -2, -3 with Myc-DEUBAD in HEK293T cells and found that over-expression of these E2s resulted in reduced DEUBAD protein levels (Fig. 6c, left panel). Interestingly, expression of UBE2Es with

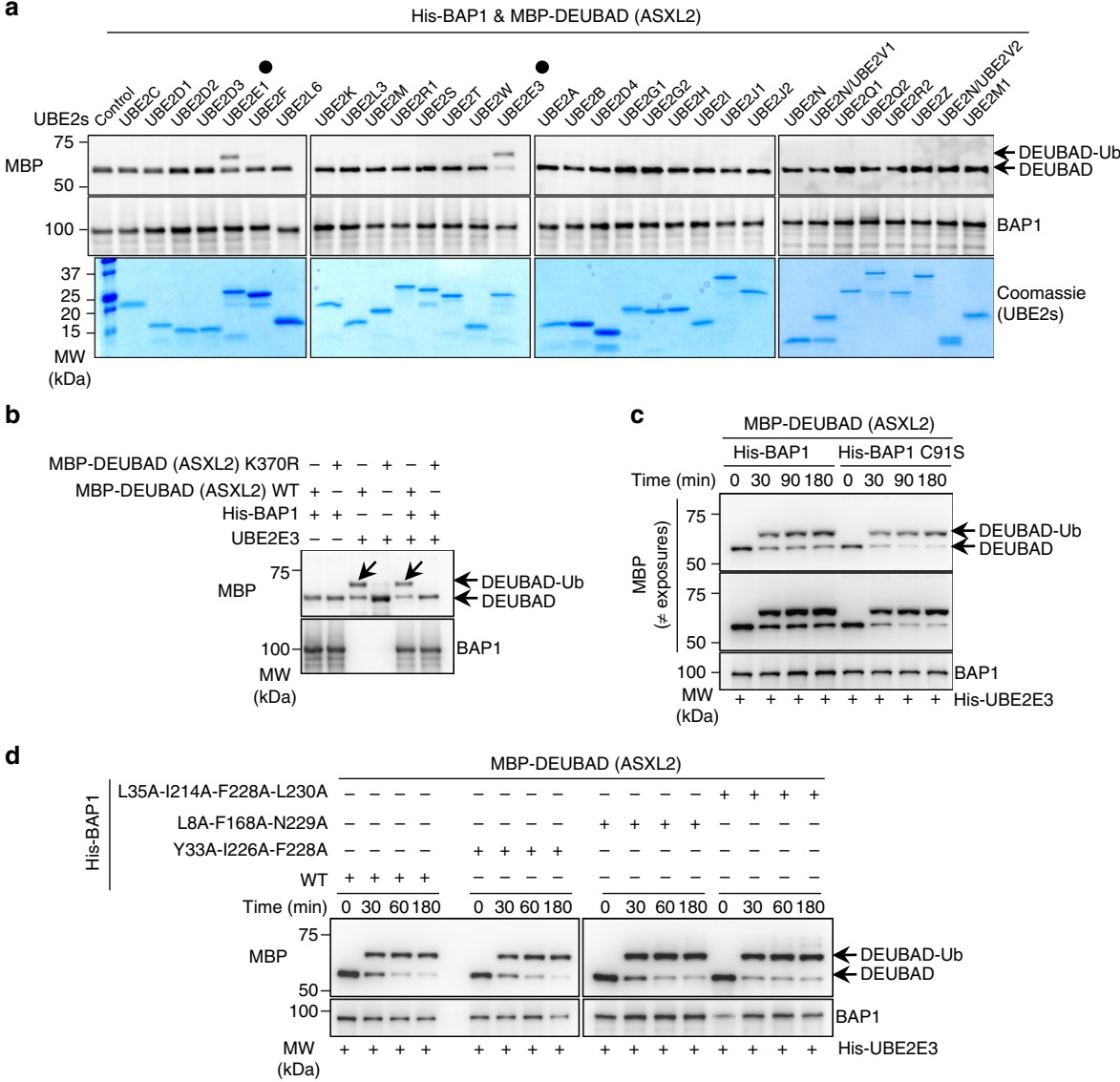

**Fig. 5** DEUBAD monoubiquitination is directly catalyzed by UBE2E family. **a** UBE2E1 and UBE2E3 directly catalyze monoubiquitination of DEUBAD (ASXL2). Bacterial purified His-BAP1/MBP-DEUBAD (ASXL2) complex was incubated with the indicated recombinant UBE2s conjugating enzymes for in vitro ubiquitination assays. The reactions were analyzed by western blot as indicated. The black dots indicate UBE2E1 and UBE2E3. $n = 1$ biological sample. **b** UBE2E3 in vitro monoubiquitinates K370 of DEUBAD (ASXL2) alone or DEUBAD (ASXL2) in complex with BAP1. Bacteria purified MBP-DEUBAD (ASXL2), MBP-DEUBAD (ASXL2) K370R, His-BAP1/MBP-DEUBAD (ASXL2) and His-BAP1/MBP-DEUBAD (ASXL2) K370R complexes were used for in vitro ubiquitination assays with bacteria-purified UBE2E3 and analyzed by immunoblotting. The arrows indicate the monoubiquitinated form of DEUBAD domain. $n = 3$ biological replicates. **c** Bacteria purified His-BAP1/MBP-DEUBAD (ASXL2) or His-BAP1 C91S/MBP-DEUBAD (ASXL2) complexes were used for in vitro ubiquitination assays. Reactions were stopped at the indicated times for immunoblotting. **d** In vitro ubiquitination assays were conducted using bacteria purified His-BAP1/MBP-DEUBAD (ASXL2) complex or MBP-DEUBAD (ASXL2) in complex with the corresponding BAP1 mutants. Reactions were done for the indicted times and analyzed by immunoblotting. $n = 2$ biological replicates for panels **c** and **d**

BAP1 resulted in decreased protein levels of the non-ubiquitinated form of the DEUBAD while a slight increase of its mono-ubiquitinated form was observed. Next, we found that UBE2Es expression had a marginal effect on full length ASXL2 protein levels (Fig. 6c, right panel). However, these enzymes significantly increased ASXL2 protein levels and notably its monoubiquitinated form following co-expression with BAP1. Since almost all endogenous ASXL2 is in complex with BAP1[33], and become constitutively monoubiquitinated (Fig. 1e and Fig. 3a, b), the primary function of ASXL2 monoubiquitination is likely to promote its stability. Accordingly, combined depletion of UBE2Es resulted in decreased protein levels of endogenous ASXL2 and BAP1

(Fig. 6d). This result suggests that, in the absence of UBE2Es-mediated monoubiquitination of ASXL2/BAP1, this complex is targeted for proteolysis through other mechanisms.

Finally, we sought to identify the *Drosophila* enzyme responsible for Asx monoubiquitination. The *Drosophila* genome encodes one single ortholog of UBE2Es called UbcD2 (Supplementary Fig. 4a). Thus, we evaluated whether knockdown of UbcD2 would affect the monoubiquitination state of Asx DEUBAD in S2 cells. We observed that dsRNA depletion of UbcD2 significantly decreased the monoubiquitinated form of DEUBAD and significantly increased the non-ubiquitinated form of the protein (Fig. 6e).

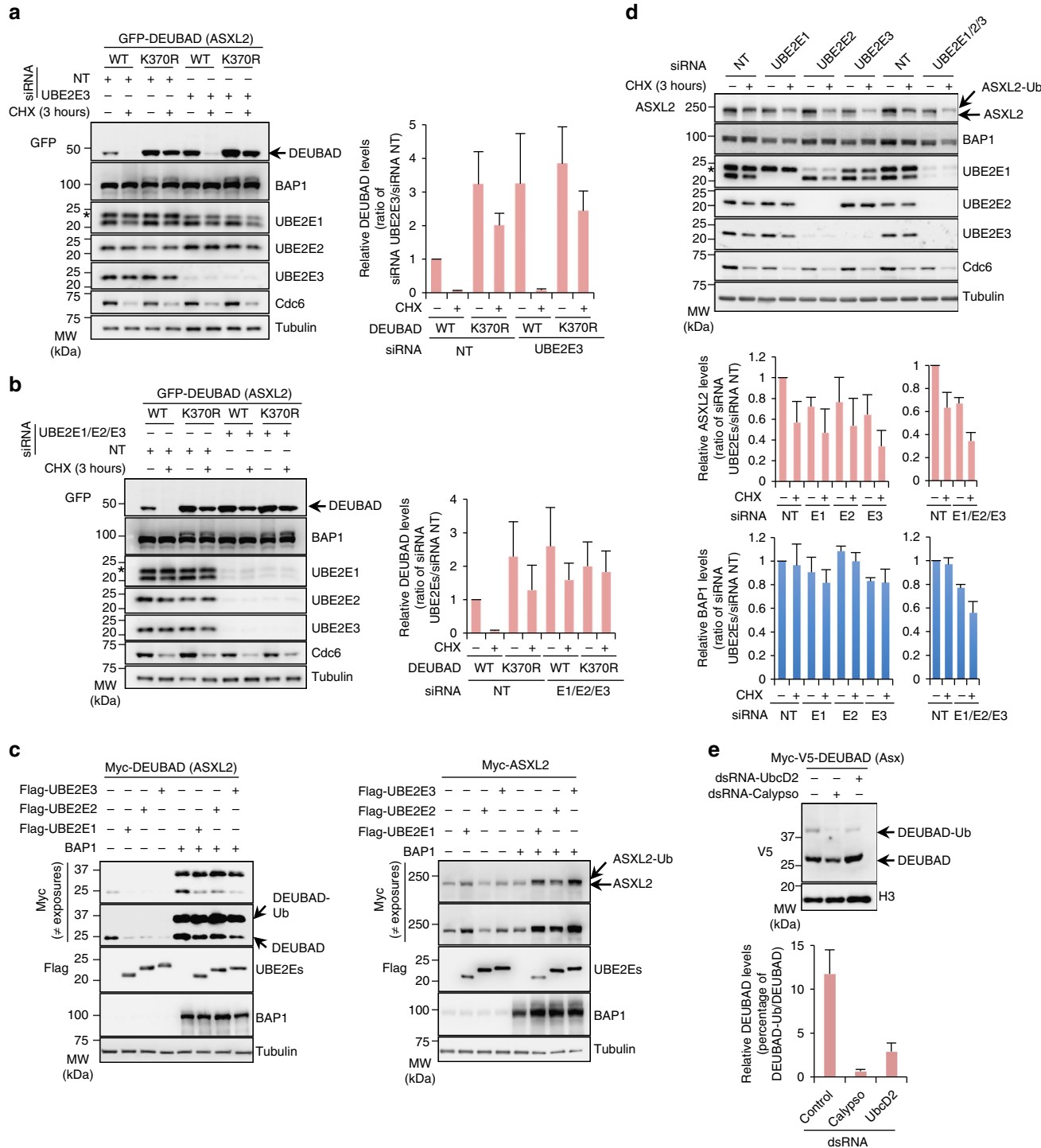

**DEUBAD monoubiquitination promotes BAP1 DUB activity.** The DEUBAD is required for stimulating BAP1 DUB activity[33,34]. Thus, we sought to evaluate the effect of DEUBAD (of ASXL2) monoubiquitination on BAP1 enzymatic activity. First, we transduced U-2 OS cells stably expressing Flag-HA-BAP1 with lentiviral expressing constructs for GFP-DEUBAD (ASXL2) or the corresponding K370R mutant form and validated that these factors are co-expressed (Supplementary Fig. 6). Of note, as expected, co-expression of BAP1 with DEUBAD or ASXL2 resulted in elevated levels of both proteins. We then conducted immunofluorescence staining for endogenous H2Aub

levels in DEUBAD- or ASXL2-expressing cells (Fig. 7a). Expression of GFP-DEUBAD, but not the K370R mutant, resulted in a strong reduction of H2Aub levels (Fig. 7a and Supplementary Fig. 7). This effect is dependent on BAP1, as expression of GFP-DEUBAD alone did not affect H2Aub levels (Fig. 7a). No reduction in H2Aub levels was observed with the catalytic dead form of BAP1 (C91S) in the presence of GFP-DEUBAD (Fig. 7b and Supplementary Fig. 7). Moreover, the same results were observed using ASXL2 since the expression of the WT form, but not the corresponding K370R mutant, induced a strong decrease of H2Aub levels (Fig. 7c). Next, we co-expressed

**Fig. 6** Dual role of UBE2E family in regulating ASXL2 protein stability. **a** Depletion of UBE2E3 results in increased DEUBAD (ASXL2) protein levels. U-2 OS cells stably expressing GFP-DEUBAD (ASXL2) were transfected with a different siRNAs oligos for *UBE2E3* and then treated with CHX as indicated. Right panel, densitometry analysis of DEUBAD protein bands was conducted and presented as indicated. $n = 3$ biological replicates. **b** Combined depletion of all three UBE2Es resulted in a stronger increase of DEUBAD (ASXL2) protein levels which become similar to those of GFP-DEUBAD K370R. U-2 OS cells stably expressing GFP-DEUBAD (ASXL2) were transfected with Non-target siRNA control or a combination of UBE2Es (*UBE2E1*, *UBE2E2*, *UBE2E3*) siRNAs, treated with CHX as indicated and used for immunoblotting. Right panel, densitometry analysis of the protein levels of DEUBAD bands were conducted and presented as in **a**. $n = 3$ biological replicates. **c** Opposite effects of UBE2Es on DEUBAD (ASXL2) and ASXL2 protein levels depending on BAP1 expression. Flag-UBE2Es expression constructs were transfected in HEK293T cells in the presence of Myc-DEUBAD (ASXL2) or Myc-ASXL2 expression vectors in presence or not of BAP1 and cells were harvested for immunoblotting. $n = 3$ biological replicates. **d** Depletion of UBE2E1, UBE2E3 and the combination of three UBE2Es resulted in decreased protein levels of endogenous ASXL2 and BAP1. U-2 OS cells were transfected with Non-target control (NT) or UBE2E siRNAs and then treated with CHX and used for immunoblotting. Bottom panel, densitometry analysis of ASXL2 and BAP1 levels was conducted and presented for each siRNA. $n = 3$ biological replicates. **e** UBE2Es-mediated DEUBAD monoubiquitination is conserved in *Drosophila*. S2 cells were transfected with dsRNA for *Calypso* or *UBCD2* and Myc-V5-DEUBAD (Asx). DEUBAD monoubiquitination was evaluated by immunoblotting. Bottom panel, densitometry analysis of the protein levels of monoubiquitinated form versus non-modified form of DEUBAD (Asx). $n = 3$ biological replicates. The stars in panels **a**, **b**, and **d** indicate UBE2E2 band detected with UBE2E1 antibody. Tubulin was used as a loading control for panels **a–d** and histone H3 was used as a loading control for panel **e**. Error bars in panels **a**, **b**, **d**, **e** represents s.d. (mean ± SD)

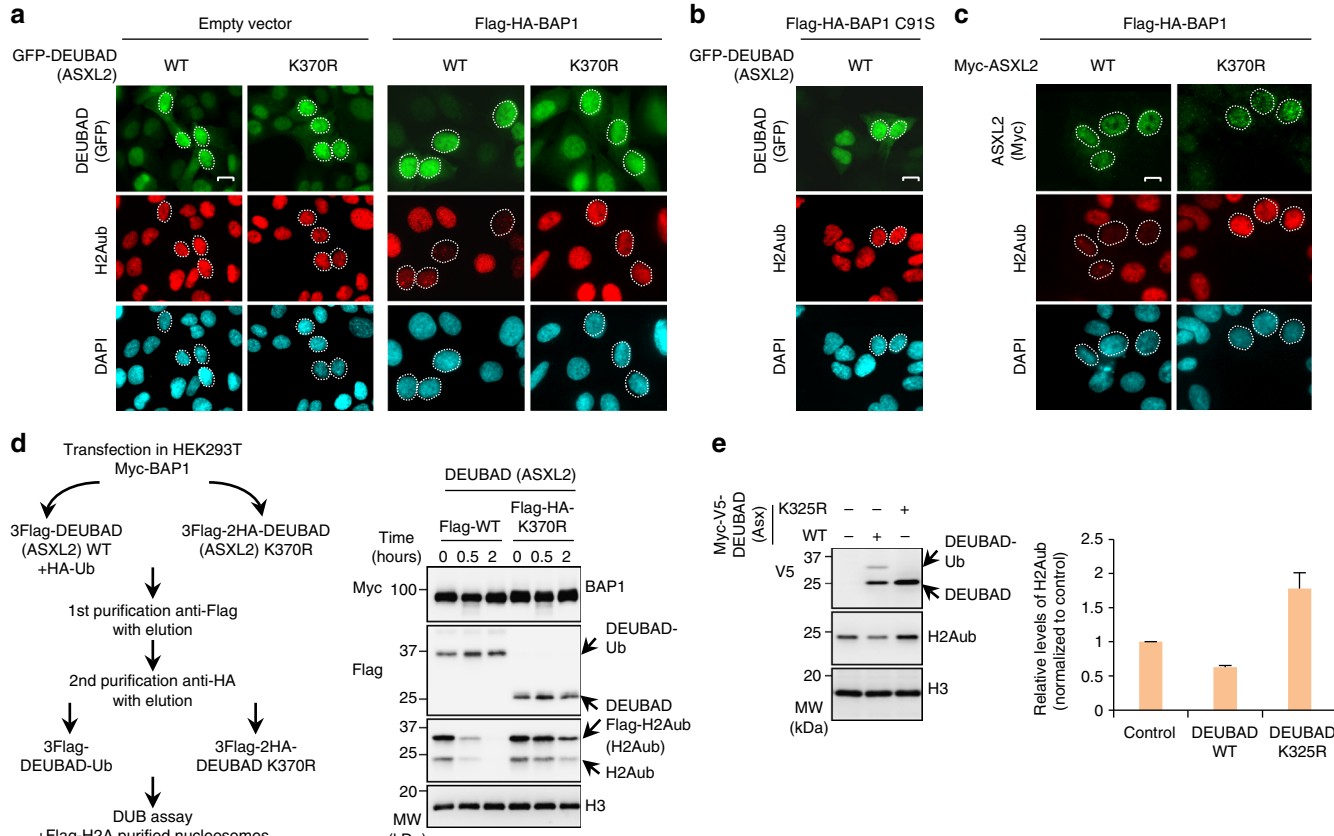

**Fig. 7** Monoubiquitination of DEUBAD K370 promotes BAP1 DUB activity. **a** Expression of DEUBAD (ASXL2) but not DEUBAD (ASXL2) K370R, results in reduced levels of H2Aub in BAP1-dependent manner. U-2 OS cells stably expressing empty vector or Flag-HA-BAP1 were transduced with lentiviral expressing vectors for either GFP-DEUBAD (ASXL2) or GFP-DEUBAD2 (ASXL2) K370R and endogenous level of H2Aub was analyzed by immunofluorescence. GFP DEUBAD (ASXL2) (green), H2Aub (red), DAPI (blue). $n = 3$ biological replicates. **b** DEUBAD induces reduction of H2Aub levels in BAP1 catalytic activity dependent manner. U-2 OS cells stably expressing Flag-HA-BAP1 C91S were infected with either GFP-DEUBAD (ASXL2) or GFP-DEUBAD (ASXL2) K370R and H2Aub levels were assessed (see also Supplementary Fig. 7). $n = 2$ biological replicates. **c** Expression of ASXL2, but not ASXL2 K370R, results in reduced levels of H2Aub. U-2 OS cells stably expressing Flag-HA-BAP1 were infected with lentiviral expressing vectors for either Myc-ASXL2 or Myc-ASXL2 K370R and H2Aub changes were evaluated by immunostaining. $n = 2$ biological replicates. The cells showing either decrease or no change of H2Aub levels were encircled in panels **a–c**. Scale bar: 10 μm for panels **a–c**. **d** Monoubiquitinated form of DEUBAD (ASXL2) strongly promotes BAP1 DUB activity comparatively to DEUBAD (ASXL2) K370R. Purified BAP1/DEUBAD complexes from HEK293T cells were used for in vitro DUB assays with Flag-H2A nucleosomes and analyzed by immunoblotting. $n = 2$ biological replicates. **e** Monoubiquitinated DEUBAD promotes deubiquitination of H2Aub in *Drosophila*. S2 cells were transfected with either Myc-V5-DEUBAD (Asx) or Myc-V5-DEUBAD (Asx) K325R and harvested for immunoblotting. Right panel, densitometry analysis of the levels of H2Aub is shown. Error bars represent s.d. (mean ± SD). $n = 3$ biological replicates. Histone H3 was used as a loading control for panels **d** and **e**

Flag-DEUBAD or Flag-HA-DEUBAD K370R with Myc-BAP1 in presence or absence of HA-Ub in HEK293T cells and purified the monoubiquitinated (Flag-DEUBAD-Ub) versus the unmodified form of DEUBAD (Flag-HA-DEUBAD K370R) in complex with BAP1 (Fig. 7d, left panel). We used these complexes for in vitro DUB assays using Flag-H2A purified native nucleosomes. Strikingly, DEUBAD-Ub strongly promoted BAP1 DUB activity towards H2Aub comparatively to the corresponding K370R mutant (Fig. 7d, right panel). As expected, based on previous studies[33,34], the unmodified DEUBAD/BAP1 complex retains a significant residual DUB activity in vitro (Fig. 7d). Remarkably, expression of the *Drosophila* Asx DEUBAD in S2 cells results in decreased H2Aub levels, while an increase of this modification was observed following expression of the corresponding K325R mutant suggesting a dominant negative effect (Fig. 7e). Thus, monoubiquitination of the DEUBAD domain of ASXL2 stimulates BAP1 DUB activity in vitro and in vivo and this effect is conserved through evolution.

**Asx K325R expression causes *Drosophila* developmental defects**. During *Drosophila* development, the homeobox transcription factor Ubx is expressed in the haltere imaginal disc but not in the wing imaginal disc. This pattern of expression is required to maintain wing and haltere identity[40]. Early studies reported that loss of function mutations of *Calypso* or *Asx* resulted in derepression of the *Ubx* locus in *Drosophila* embryos and imaginal discs[16,41]. Thus, we set up experiments to evaluate the impact of over-expressing Asx K325R on Ubx expression and *Drosophila* development. We analyzed H2Aub levels as an indicator of Calypso DUB activity. We used the UAS/Gal4 system to express transgenes during larval wing and haltere development[42]. The engrailed-Gal4 (en-Gal4) driver directs transgene expression in the posterior compartment of the larval wing, haltere and leg discs (GFP expression in Fig. 8a and Supplementary Fig. 8). Strikingly, while expression of Asx promoted deubiquitination of H2Aub (Fig. 8a, 1′, 2′), expression of Asx K325R mutant led to increased levels of H2Aub, suggesting that it dominantly interfere with the catalytic activity of the endogenous Calypso/Asx complex (Fig. 8a, 3′, 4′). We note that expression of Asx K325R mutation resulted in a discontinuous expression of GFP in the discs. This effect does not seem to be the consequence of ectopic cell death as the signal for activated Caspase-3 remained essentially similar between WT and mutant *Asx* (Supplementary Fig. 8c, 3′, 4′). Overexpression of the *Drosophila* tumor necrosis factor (TNF), Eiger, was performed as a positive control for Caspase-3 staining[43,44] (Supplementary Fig. 8c, 2, 2′). We also note that WT and K325R *Asx* transgenes are expressed at comparable levels (Supplementary Fig. 8a, b). We did not observe a noticeable effect of Asx K325R on *Ubx* expression in wing discs (Fig. 8a, 3″). However, expression of Asx K325R resulted in a significant repression of Ubx in haltere discs (Fig. 8a, 4″), and this was accompanied by a partial haltere to wing homeotic transformation (Fig. 8b). Moreover, adult *Drosophila* expressing Asx K325R mutant had a crumpled wing phenotype (Fig. 8c). Taken together, our results suggest that the Asx K325R mutant exerts a dominant negative effect over the endogenous Asx leading to increased levels of H2Aub and developmental defects in *Drosophila*.

**Monoubiquitination of ASXL2 regulates cell proliferation**. As the BAP1/ASXL2 complex regulate cell proliferation[33], we sought to determine the requirement of ASXL2 monoubiquitination in this process. First, we conducted cell cycle synchronization and found that the DEUBAD is constitutively modified in all cell cycle phases (Supplementary Fig. 9a, b). Next, we used lentiviral constructs with ASXL2 and corresponding K370R mutant, and transduced different model cell lines. As ASXL2 K370R is in general more stable than the WT, we used various virus titers in order to compare the biological effects of ASXL2 and the corresponding mutant with similar levels of protein expression. In U-2 OS cells, expression of ASXL2 K370R resulted in reduced cell proliferation as determined by colony forming ability assays (Fig. 9a). Next, we used nocodazole to arrest cells in M phase and prevent cell cycle re-entry, thus allowing us to observe their progression from G1 to M phases. While the majority of U-2 OS cell population transduced with empty vector or ASXL2 progressed to M phase, cells expressing the ASXL2 K370R were significantly delayed in their progression from G1 to M (Fig. 9b). Expression of ASXL2 K370R also resulted in decreased cell proliferation in IMR90 normal human fibroblasts (Fig. 9c, d). We also tested additional cell types notably LF1 skin fibroblast and 3T3L1 mouse preadipocytes and observed similar phenotypes (Supplementary Fig. 9c-f). These results predict that depletion of ASXL2 itself would phenocopy the overexpression of ASXL2 K370R. Hence, we depleted ASXL2 in U-2 OS by siRNA and observed reduced colony forming ability and cell cycle progression from G1 to M following nocodazole block (Fig. 9e). Consistent with these observations, depletion of BAP1 in U-2 OS also resulted in reduced colony forming ability and delayed G1 to M cell cycle progression (Supplementary Fig. 10a, b). We also conducted a double thymidine block to arrest U-2 OS cells at the G1/S barrier, and observed a slower progression of both BAP1- and ASXL2-depleted cells following release from cell cycle arrest (Supplementary Fig. 10c). To further support our observations that cell cycle defects are directly linked to ASXL2 monoubiquitination, we sought to determine the impact of one of the main drivers of this modification, notably UBE2E3, on cell proliferation. First, we depleted UBE2E3 in U-2 OS cells by siRNA and observed reduced cell proliferation (Fig. 9f). Next, we generated *UBE2E3* knockout cells using CRISPR/Cas9 and observed reduced colony forming ability (Fig. 9g, h). In addition, we observed a delayed cell cycle progression of UBE2E3-deficient cells following nocodazole treatment, as about half population of *UBE2E3* KO cells remained in G1, while the majority of control cells progressed to M (Fig. 9i). Based on these results, we concluded that monoubiquitination of ASXL2 DEUBAD is critical for mammalian cell proliferation.

**Expression levels of BAP1, ASXL2, and UBE2E3 in mesothelioma**. Since we found that BAP1 activity is tightly regulated by UBE2Es-mediated ASXL2 monoubiquitination, we analyzed protein expression of these proteins in malignant mesothelioma, a cancer characterized by a high frequency of BAP1 mutations[45,46]. We reasoned that as these factors act within a signaling pathway that coordinate BAP1 DUB activity, their expression should be co-regulated. Changes of UBE2Es expression is expected to impact BAP1 and ASXL2 expression. In turn, loss of BAP1 function by inactivating mutations, might also impact the expression of UBE2Es, as part of a potential feedback loop. We studied BAP1, ASXL2, and UBE2Es expression in 10 human malignant mesothelioma biopsies containing wild-type *BAP1* and in 10 human mesothelioma biopsies containing mutated *BAP1*, as determined in previous studies[46,47]. We quantified positive readings as 1.0 and negative readings as 0.0, with gradations in between (Supplementary Table 3). Applying a logistic regression model with BAP1 status (WT vs. mutant) as the outcome, the best predictors were UBE2E3 ($p = 0.04$) and ASXL2 ($p = 0.04$). We calculated correlations among UBE2E1, UBE2E2, UBE2E3, and ASXL2 ($N = 20$). The strongest association was between

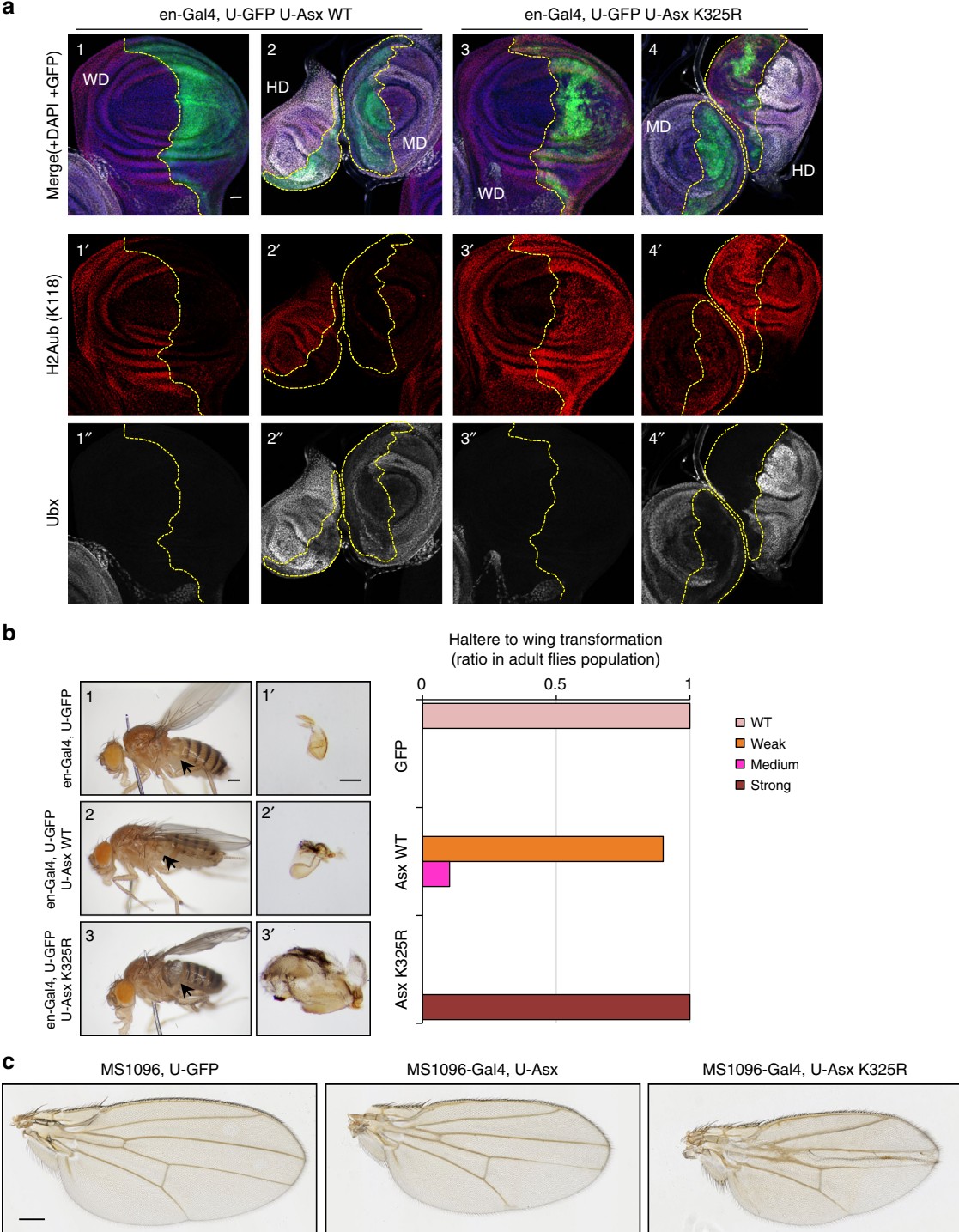

**Fig. 8** Expression of Asx K325R results in haltere to wing transformation. **a** Effect of the expression of Asx WT or K325R mutant during *Drosophila* larval disc development. Discs were stained with anti-H2Aub (red), anti-Ubx (white) and DAPI (blue). GFP (green) indicates expression pattern of *en-Gal4* and is delimited by dashed lines. Genotypes are indicated (see also Supplementary Fig. 8). Scale bar: 20 μm. *n* = 3 biological replicates. **b** Partial haltere to wing homeotic transformation associated with expression of Asx K325R. Phenotypes associated with the expression of Asx K325R in adult *Drosophila*. Left panel, images showing whole *Drosophila* animals and dissected halteres. Right panel, quantification of haltere to wing transformation phenotype. The arrows indicate the halteres. *n* = 3 biological replicates. Scale bar: 200 μm for images 1, 2 and 3. Scale bar: 100 μm for images 1′, 2′ and 3′. **c** Wing development defects following expression of Asx or Asx K325R. Scale bar: 200 μm. *n* = 3 biological replicates. WD wing imaginal disc, HD Haltere imaginal disc, MD mesothoracic leg imaginal disc

UBE2E2 and ASXL2 ($r = 0.55$, $p = 0.01$), followed by UBE2E2 and UBE2E3 ($r = 0.49$, $p = .03$), and UBE2E3 and ASXL2 ($r = 0.32$, $p = .17$). When restricting the analysis to wild-type only ($N = 10$), the strongest association was still between UBE2E2 and ASXL2 ($r = 0.90$, $p = 0.0004$), followed by UBE2E2 and UBE2E3 ($r = 0.50$, $p = 0.14$), and UBE2E3 and ASXL2 ($r = 0.48$, $p = 0.16$) (Fig. 9j and Supplementary Fig. 11a). These results suggest that BAP1, ASXL2, and UBE2Es expression are correlated in

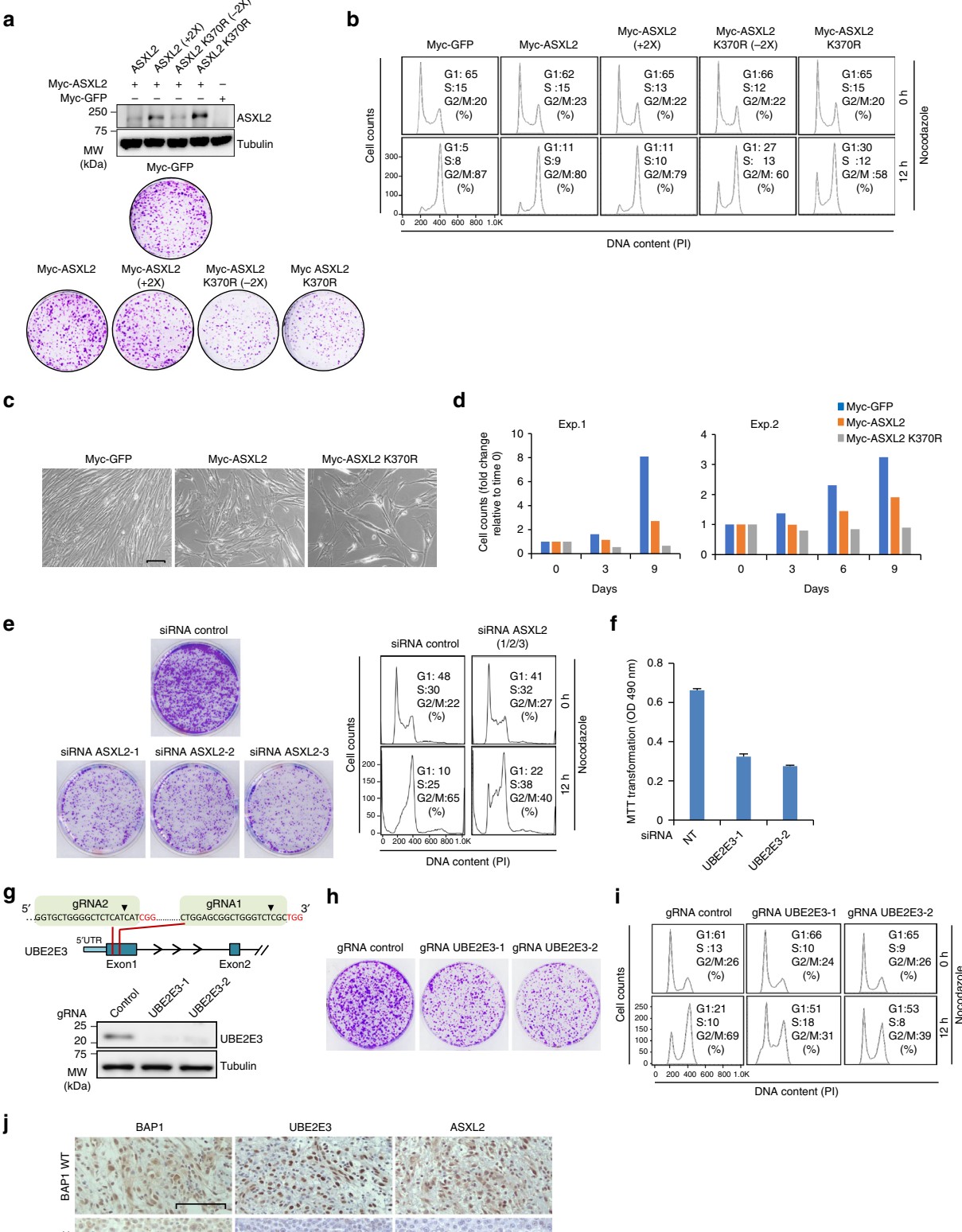

mesothelioma tissues and are in further support of the functional interaction between UBE2Es and the BAP1/ASXL2 complex.

## Discussion

In this study, we provide an example of a highly conserved cross-regulation between a DUB and its cofactors (Fig. 10). Remarkably, ubiquitination targets a highly conserved lysine within the

DEUBAD domain which is used for interaction with BAP1. Thus, it is conceivable to postulate that this monoubiquitination event might induce conformational changes in BAP1/DEUBAD in order to increase the affinity of this complex for substrates. Whether this is actually the case and whether monoubiquitination of DEUBAD is also required for deubiquitination of other substrates, including BAP1 self-deubiquitination we previously established[14], remains interesting questions to investigate. On the

**Fig. 9** Expression of ASXL2 K370R reduces mammalian cell proliferation. **a** Enforced expression of ASXL2 K370R decreases cellular proliferation. U-2 OS cells were transduced with different amounts of lentiviral suspensions produced using ASXL2 or ASXL2 K370R constructs. Cells were selected by puromycin and harvested for immunoblotting (top panel). Equal numbers of puromycin-selected cells were plated for colony formation assay (CFA) (bottom panel). $n = 2$ biological replicates. **b** The cells infected in **a**, were treated with nocodazole for FACS analysis at the indicated times. Note that ($+2\times$) refers to transduction of cells with twice the amount of virus we normally use for Myc-ASXL2, and ($-2\times$) refers to transduction of the cells with two times less the amount of viruses we normally use for ASXL2 370R. This adjustment was conducted to correct for the expression levels usually higher for ASXL2 K370R. $n = 2$ biological replicates. **c**, **d** Normal diploid fibroblast IMR90 cells were transduced with viral expression constructs for ASXL2 or ASXL2 K370R. Cells were selected by puromycin and equal numbers were plated for phase contrast pictures (**c**) or cell counts (**d**). Scale bar: 50 μm for panel **c**. $n = 2$ biological replicates. (Exp.1 and Exp.2). **e** siRNA depletion of ASXL2 decreases cellular proliferation. U-2 OS cells were transfected with NT siRNA control or siRNA for *ASXL2*. Equal numbers of puromycin-selected cells were plated for CFA (left panel). Cells were treated with nocodazole for FACS analysis (right panel). **f** siRNA depletion of UBE2E3 decreases cellular proliferation. U-2 OS cells were transfected with individual siRNA constructs as indicated. Cells were plated for viability measurement using MTT assay. $n = 3$ biological replicates. Error bars represent s.d. (mean ± SD). **g**, **h** Inactivation of *UBE2E3* locus decreases cellular proliferation. Schematic representation for gRNAs targeting the UBE2E3 locus (**g** top panel). U-2 OS cells were transduced with different lentiviral CRISPR/Cas9 constructs, selected by puromycin and harvested for immunoblotting (**g** bottom panel). $n = 3$ biological replicates. Equal numbers of puromycin-selected cells were plated for CFA (**h**). $n = 2$ biological replicates. **i** The cells selected as in **h** were treated with nocodazole for FACS analysis at the indicated time . $n = 2$ biological replicates. **j** Positive correlation of BAP1, ASXL2, and UBE2E3 protein expression levels in human mesothelioma. Mesothelioma biopsies were immunostained for ASXL2, UBE2E3, or BAP1 (see Supplementary Fig. 11a and Supplementary Table 3). Pictures were taken at 100× magnification. Scale bar: 100 μm. Tubulin was used as a loading control for panels **a** and **g**

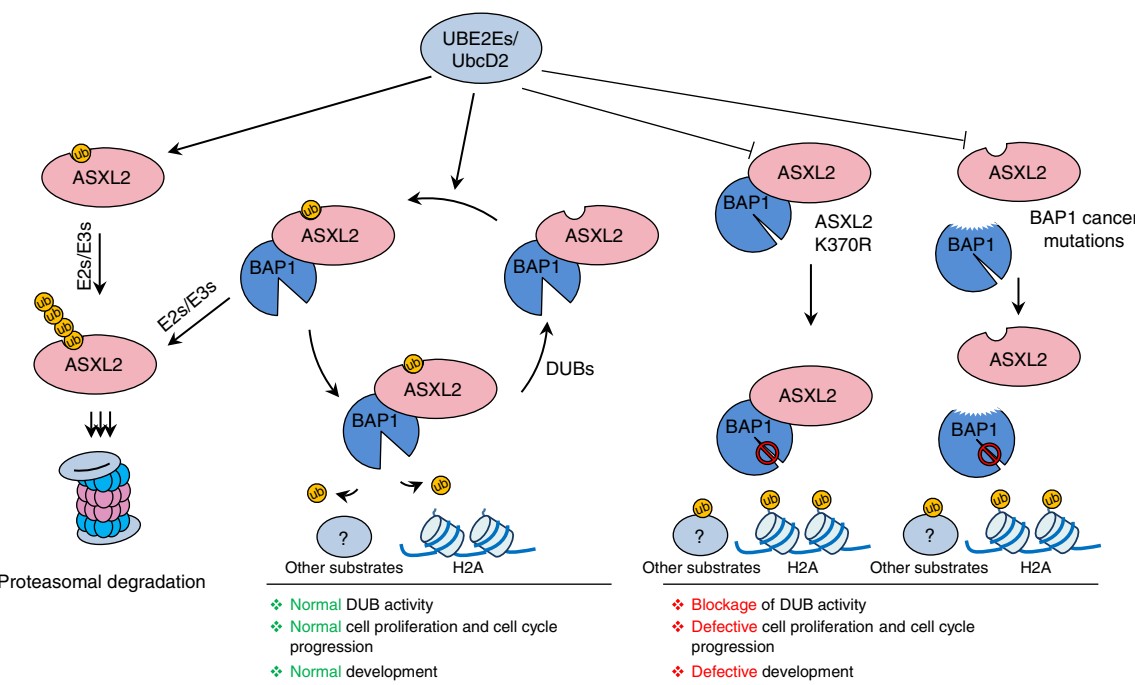

**Fig. 10** Regulation of the BAP1/ASXL2 complexes by DEUBAD ubiquitination. ASXL2 is constitutively monoubiquitinated by UBE2Es on its DEUBAD domain. Interaction of monoubiquitinated ASXL2 with BAP1 leads to its stabilization and the subsequent activation of the DUB complex. Otherwise monoubiquitinated ASXL2 is targeted by other E3 Ub-ligases for ubiquitin chain extension and proteasomal degradation. UBE2Es also target ASXL2 already in complex with BAP1, possibly dynamically regulating its activity. DUBs might also regulate the stability of both free and complexed ASXL2. Mutation of ASXL2 lysine 370 or cancer mutations that abolish ASXL2-BAP1 interaction lead to defective monoubiquitination and subsequent loss of BAP1 DUB activity and tumor suppression

other hand, it is possible that the monoubiquitination of ASXL2 recruits other interacting partners to the BAP1/ASXL2 complex to enhance either substrate recognition or recruitment.

We showed that nearly all the cellular pool of endogenous ASXL2 is monoubiquitinated. As most nuclear pool of ASXL2 is in complex with BAP1[33], our data suggest that the very small fraction of the non-ubiquitinated ASXL2 is likely to correspond to the free form of the protein. We propose a model whereby free ASXL2, e.g., newly synthesized protein, is very unstable and following its monoubiquitination, subsequent Ub chain extension on the same site induces its proteasomal degradation. The interaction with BAP1 would play an important protective role against the eventual action of cellular E3 ligases that might

induce Ub chain extension and proteasomal degradation. Consistent with this model, ubiquitination of lysine 370 of ASXL2 plays a dual role in promoting protein stability (in the presence of interaction with BAP1) or inducing protein degradation (in the absence of interaction with BAP1). Indeed, mutation of K370 results in increased protein levels of DEUBAD or ASXL2, and expression of chain elongation-defective Ub mutants, as well as the inhibition of the proteasome-associated DUB PSMD14, result in both stronger monoubiquitination and the apparition of additional ubiquitinated species of DEUBAD.

Interestingly, we found that UB2E2s catalyze mono-ubiquitination of DEUBAD, without requiring any E3 Ub-ligase in vitro. Consistent with DEUBAD K370 mutagenesis studies, we

found that expression of UBE2E enzymes, which promote ASXL2 monoubiquitination on K370, lead to either DEUBAD stabilization or proteolysis. These effects were quite remarkable and are likely due to high levels of expression and rapid turnover of the DEUBAD domain. In contrast, UBE2Es expression did not have a noticeable effect on ASXL2 alone, but increased its monoubiquitination and protein levels following co-expression with BAP1. These results suggest that, in the absence of ASXL2 monoubiquitination by UBE2Es, other mechanisms downregulate ASXL2/BAP1, preventing a possible dominant negative effect of a catalytically inactive DUB complex. Thus, UBE2Es ensure a rapid turnover of free ASXL2 while stabilizing ASXL2 integrated within the BAP1 complex, hence tightly regulating the dosage of ASXL2 and DUB activity. This might constitute an important quality control mechanism for ASXL2/BAP1 protein levels and DUB activity. Excess of free ASXL2 protein might be quickly primed by UBE2Es for degradation to prevent potential unwanted effects of orphan ASXL2. As we demonstrated that monoubiquitination of DEUBAD is promoted by the assembly of the composite ubiquitin-binding interface, which requires proper BAP1 folding, we also concluded that monoubiquitination of DEUBAD could provide a quality control mechanism for the assembly of a DUB activity competent complex.

While we observed a robust monoubiquitination of DEUBAD by UBE2E enzymes, in vitro, in the absence of a typical E3 ligase, we cannot exclude at this point that a yet to be identified E3 ligase is required for efficient monoubiquitination in vivo. For instance, this highly specific site monoubiquitination of DEUBAD is reminiscent of SUMOylation which could be catalyzed by UBC9 E2 only, in vitro[48–50], but yet directed by SUMO E3 ligases in vivo[51–53]. The molecular mechanisms that orchestrate UBE2E family enzymes are not fully understood, but, in line with our data indicating that UBE2E enzymes catalyze monoubiquitination of ASXL2, a recent study shows that these enzymes have a distinct N-terminal extension that limits Ub chain assembly on substrates[54]. Thus, other potential E3s promoting ASXL2 degradation are likely to act in concert with UBE2Es. On the other hand, it is also possible that polyubiquitination of ASXL2 might be ensured by other E2 conjugating enzymes and E3 ligases independently of UBE2E family. Consistent with this idea, mutation of K370 of ASXL2 partially inhibits ASXL2 turnover and we identified other ubiquitination sites on this factor. These results also suggest that ubiquitination plays a pervasive role in the regulation of ASXL2 function. We also noticed that depletion of several DUBs lead to the reduction of DEUBAD protein levels. It will be highly interesting to identify which DUBs directly and specifically targets the monoubiquitinated form and/or preventing Ub chain assembly and degradation of ASXL2. As ASXL2 is constitutively monoubiquitinated, it will be interesting to determine how this complex is protected from deubiquitination or Ub chain elongation and subsequent proteasomal degradation. Also, an important question is what are the signals that can switch monoubiquitinated ASXL2 to its polyubiquitinated form resulting in its proteasomal degradation.

*Drosophila* Asx possesses both polycomb and trithorax activities, i.e., promoting gene transcription silencing or activation respectively[18]. However, we did not observe a noticeable derepression of *Ubx* in wing imaginal discs. Thus, although *Asx* K325R appears to act as a dominant-negative allele on the endogenous DUB complex, its effect is possibly not strong enough to displace all endogenous complexes repressing *Ubx* in wing imaginal discs. The crumpled wing phenotype is likely to be caused by a slight dominant negative effect of Asx K325R on endogenous Asx. In contrast, expression of Asx K325R in haltere imaginal discs in which *Ubx* is already active results in a noticeable repression of this gene and a partial haltere to wing

homeotic transformation, consistent with the role of *Ubx* as a determinant of cell identity that precludes development of haltere imaginal discs into wings. Thus, it is possible that *Ubx* active gene transcription might be more permissive to the dominant negative effect of Asx K325R than *Ubx* repressed state. It is also interesting to note that expression of the Asx K325R mutant might interfere with the *en-Gal4* gene itself and this might explain the patchy expression of GFP in the discs. While this will require other studies, our results, nonetheless, show that expression of Asx K325R resulted in distinct *Drosophila* phenotypes suggesting the importance of Asx monoubiquitination in vivo.

BAP1 regulates cell cycle progression[8,55] and we recently found that BAP1 interaction with ASXLs is important for normal cell proliferation[33]. Here, we found that depletion of E2 enzymes responsible for ASXL2 monoubiquitination as well as expression of monoubiquitination-defective ASXL2 mutant reduces cell proliferation in several cell types suggesting the importance of this modification for normal cell cycle progression. Although H2Aub was previously involved in cell proliferation[56], it is not clear, at this time, whether the observed effects of ASXL2/BAP1 defective in monoubiquitination are solely due to disruption of H2A deubiquitination or the effect of BAP1/ASXL complexes on other substrates.

Our data suggest that the signaling pathway we characterized here is important for tumor suppression. Indeed, a cancer mutation of BAP1 CTD found in melanoma[33], that abrogate BAP1-ASXL2 interaction and DUB activity toward histone H2Aub results in highly reduced DEUBAD monoubiquitination. In addition, all BAP1 domains and motifs, including UCH, CC1, and CTD involved in catalysis as well as DEUBAD domains of ASXLs are targeted by cancer mutations (Supplementary Fig. 11b, c), Finally, we found that a positive BAP1 nuclear staining in mesothelioma tumors, indicative of wild-type BAP1[46], correlated with positive staining for ASXL2 and UBE2Es, particularly UBE2E2 and UBE2E3, while negative nuclear staining, indicative of BAP1 mutation correlated with negative or reduced (focal) staining for these same proteins. These findings suggest that deregulation of the UBE2E-ASXL2/BAP1 Ub signaling axis might participate to mesothelioma development.

## Methods

**Cell lines.** Primary human skin fibroblasts (LF1) (ATCC, PCS-201-013), U-2 OS osteosarcoma (ATCC, HTB-96), human embryonic kidney HEK293T (293T) (ATCC, CRL-3216), normal Human Lung Fibroblasts (IMR90) (ATCC, CCL-186), and 3T3-L1 mouse preadipocytes (ATCC, CL-173) were cultured in Dulbecco's modified Eagle's medium (DMEM) supplemented with 10% of fetal bovine serum (FBS), 1% L-glutamine, and 1% penicillin/streptomycin. *D. melanogaster* cell line S2 were cultured in Schneider medium (Sigma) supplemented with 10 % FBS. All the cell lines have been continuously tested negative for mycoplasma contamination using DAPI staining. Mammalian cell lines were authenticated from ATCC. After obtaining them, many batches were frozen and proper cell morphology and proliferation rates were continuously checked. None of the cell lines cited above are listed in the database of commonly misidentified cell lines, ICLAC.

**Bacterial strains.** TOP10 chemically competent E.Coli bacteria (One Shot™ TOP10 Chemically Competent E. coli) were purchased from ThermoFisher (C404010). Rosetta™ 2(DE3) Competent bacteria were obtained from Novagen (73197). BL21 CodonPlus-RIL bacteria were obtained from Agilent (230240). All bacteria were stored at −80 °C and grown in LB medium at 37 °C. TOP10 bacteria were mainly used for plasmids amplification and cloning procedures. Rosetta and BL21 bacteria were used for protein recombinant expression and purifications.

**Plasmids.** pENTR ASXL2, pENTR ASXL2ΔDEUBAD, pENTR DEUBAD (ASXL2), pENTR DEUBAD (ASXL1), Myc-ASXL2, Myc-ASXL2ΔDEUBAD, MBP-DEUBAD (ASXL2) WT, and Flag-BAP1ΔR666-H669 were reported[33]. pET30a + BAP1 for production of His-tagged BAP1 was also described[10]. Flag-BAP1, Flag-BAP1ΔUCH, Flag-BAP1ΔCC1, Flag-BAP1ΔHBM, Flag-BAP1ΔCTD, GFP-UCH, GFP-UCH-CC1, Myc-CTD, non-tagged pCDNA3-BAP1, pCDNA3-BAP1-C91S, HA-Ub, Myc-UBE2O, and Myc-UBE2O CD were also reported[14]. Myc-CTD ΔR666-H669 was generated by gene synthesis (Biobasic Inc.). DEUBAD

(ASXL2) (K289R, K340R, K365R, K370R, K390R, K391R) and DEUBAD (ASXL1) (K351R, K353R, K362R) point mutations constructs were generated by site direct mutagenesis using Q5 DNA Polymerase (CAT#M0491L) (New England Biolabs) in pENTR D-Topo DEUBAD (ASXL2) and pENTR D-Topo DEUBAD (ASXL1) respectively. Mammalian expression vectors of DEUBAD (ASXL1) and DEUBAD (ASXL2) mutant forms were generated in pDEST-Myc vector by recombination using LR clonase kit (CAT#11791020) (ThermoFisher Scientific). MBP-DEUBAD (ASXL2) K370R was generated by recombination of pENTR D-Topo DEUBAD (ASXL2) K370R into pDEST-MBP vector. DEUBAD (ASXL3) 240-280 a.a and DEUBAD (ASXL3) K350R 240-280 a.a were generated by gene synthesis (Biobasic Inc.) and then subcloned into pENTR D-Topo vector. Myc-DEUBAD (ASXL3) and Myc-DEUBAD (ASXL3) K350R were generated by recombination of pENTR D-Topo DEUBAD (ASXL2) vectors into pDEST-Myc vector. pENTR ASXL2Δ-DEUBAD and pENTR ASXL2ΔPHD were generated by PCR-based subcloning using pENTR ASXL2 as a template cDNA. pENTR ASXL2 K370R was generated by site directed mutagenesis using Q5 DNA Polymerase followed by subcloning into pENTR ASXL2. HA-ASXL2 WT and HA-ASXL2 K370R were generated by subcloning each cDNA into a modified HA-pENTR D-Topo plasmid. GFP-DEUBAD (ASXL2) WT and GFP-DEUBAD (ASXL2) K370R were generated by subcloning DEUBAD (ASXL2) into a modified GFP-pENTR D-Topo plasmid. The mammalian expression vector of HA-ASXL2 WT, HA-ASXL2 K370R, GFP-DEUBAD(ASXL2) WT, and GFP-DEUBAD(ASXL2) K370R were generated by recombination into either puromycin-resistant (for HA constructs) or blasticidin-resistant (for GFP constructs) lentiviral pLenti-CMV expression vector (Addgene #17452 and # 17451 for puro and blast resistance respectively). 3xFlag-DEUBAD (ASXL2) WT and 3xFlag-2HA-DEUBAD (ASXL2) K370R were generated by subcloning of DEUBAD (ASXL2) WT and DEUBAD (ASXL2) K370R cDNA into pLPC vector. pRK5-HA-Ubiquitin-K48, pRK5-HA-Ubiquitin-K48R, pRK5-HA-Ubiquitin-K63, pRK5-HA-Ubiquitin-K0, and pCDNA-HA-UCH37 were purchased from Addgene (#17605, #17604, #17606, #17603, and # 19415). BAP1 Ub binding mutants, Y33A/I226A/F228A, L8A/F168A/N229A, and L35A/I214A/F228A/L230A were generated by gene synthesis (BioBasic) and then subcloned into a modified pENTR D-Topo plasmid (Life Technologies). Mammalian and bacterial expression constructs of Flag-BAP1 mutants were generated by recombination of pENTR BAP1 into pDEST-Flag or pDEST-His expression vectors respectively. pENTR-Myc constructs of ASXL2 WT and ASXL2 K370R were generated by subcloning each respective cDNA from pENTR to a modified Myc-pENTR D-Topo plasmid. Lentiviral expression constructs of ASXL2 WT and ASXL2 K370R were generated by recombination into pLenti-CMV expression vector. UBE2E1, UBE2E2 and UBE2E3 cDNA were produced by gene synthesis (BioBasic Inc.) and subcloned into modified pENTR D-Topo. Mammalian and bacterial expression vector for UBE2E1, UBE2E2, and UBE2E3 were generated respectively by recombination into pDEST-Flag and pDEST-His. DEUBAD (RPN13) 250-407 a.a and DEUBAD (INO80G) 1-180 a.a were amplified by PCR from U-2 OS cDNA using Q5 DNA Polymerase followed by subcloning into pENTR D-Topo. Mammalian expression vector of DEUBAD (RPN13) and DEUBAD (INO80G) were generated by recombination of their respective pENTR D-Topo DEUBAD vectors into pDEST-Myc. pENTR-DEUBAD (Asx) WT was generated by gene synthesis of the *Drosophila* DEUBAD (Asx) domain (185-352aa) (ThermoFisher Scientific) into modified pENTR D-Topo. Calypso full-length cDNA was generated by gene synthesis (BioBasic Inc.) into modified pENTR D-Topo. pENTR-DEUBAD (Asx) K325R was generated by site-directed mutagenesis using Q5 High-Fidelity DNA Polymerase into pENTR-DEUBAD (Asx) WT. Myc-V5-DEUBAD (Asx) WT, Myc-V5-DEUBAD (Asx) K325R and 3xFlag-Calypso WT were generated by PCR using their respective pENTR vectors as templates and then subcloned into pActin5C vector for ubiquitous expression in S2 cells. Asx full length was generated by gene synthesis (BioBasic Inc.) and sub-cloned into pACT-V5-Myc and pUASAttb plasmids. Plasmids were verified by DNA sequencing. psPAX2 (Addgene # 12260) and pMD2-G (Addgene # 12259) were used for lentivirus packaging. pLentiCRISPR_V2 vector for gRNA cloning was purchased from Addgene (#52961).

**Chemical and reagents**. Ub Activating Enzyme (UBE1) (E-305), Ub-AMC (CAT# U-550), Ub (CAT#UM-I44A), Ub-Vinyl Methyl Ester (Ub-VME) (CAT#U-203), and UBE2 conjugating enzymes (Supplementary Table 6) were from Boston Biochem. Cycloheximide (CHX) (CAT#C1988), MG132 (CAT#C2211), Thymidine (CAT # T9250), Micrococcal nuclease (MNase) (CAT# N3755) and Schneider's Insect Medium (CAT#S0146) were from Millipore SIGMA and polyethylenimine (PEI) (CAT#23966-1) was from Polysciences Inc. *N*-methylmaleimide (NEM) (CAT#ETM222), was from Bioshop, CDK1 inhibitor RO-3306 (10 μM) (CAT# 217699) and Nocodazole (CAT#487928) where purchased from Millipore. Effectene transfection kit (CAT#301427) was from Qiagen. RNAimax (CAT# 13778150) was from ThermoFisher Scientific. IPTG (CAT#IPT002) was from Bioshop. MTT (3-(4,5-Dimethylthiazol-2-yl)-2,5-diphenyltetrazolium bromide; CAT#M5655) was from Sigma-Aldrich.

**Recombinant protein production**. Bacterial induction and protein purification procedures were conducted as follow[33]. The protein expression constructs were transformed into Rosetta or BL21 bacteria. For protein induction, the cells were grown at 37 °C and then treated, at exponential growth phase, with IPTG at 0.4 mM

at 25 °C to induce protein expression. Cells were harvested in cold PBS and then bacterial pellet was collected by centrifugation at 4000 r.p.m. for 20 min. Recombinant proteins were purified under native conditions. Cell pellets were lysed in 50 mM Tris-HCl pH 8, 500 mM NaCl and 3 mM DTT, 1 mM PMSF, 1X protease inhibitors, and left on ice for 30 min. After incubation, suspensions were sonicated and centrifuged at 16,000 r.p.m. for 20 min. Supernatants were incubated with Ni-NTA Agarose resin (Invitrogen #R901–15) for His-BAP1, His-BAP1 C91S and His-UBE2Es or with Amylose resin (New Englands Biolabs, CAT#E8021) for MBP-DEUBAD (ASXL2) and MBP-DEUBAD (ASXL2) K370R overnight at 4 °C. For His-purified proteins, the resin was then washed 6 times with 20 volumes of 50 mM Tris-HCl pH 8.0, 500 mM NaCl, 3 mM DTT and 20 mM imidazole and transferred into a Bio-Spin Disposable Chromatography columns (Bio Rad #732–6008). Proteins were eluted with 200 mM imidazole and loaded on SDS-PAGE for Coomassie brilliant blue staining using BSA for relative quantification. Amylose resin were washed with 50 mM Tris pH 7,3; 300 mM NaCl; 1 % Triton; 5 mM DTT; 1 mM EDTA; 1 mM PMSF and protease inhibitors cocktails. The purified MBP-DEUBAD (ASXL2) proteins were kept on the amylose agarose beads in 50 mM HEPES pH 8.0; 50 mM NaCl; 10 % Glycerol and 1 mM DTT for in vitro ubiquitination assays.

**Cell transfection**. HEK293T cells were transfected with the mammalian expressing vectors using PEI. Three days post-transfection, cells were harvested for immunoblotting or immunoprecipitation. U-2 OS, IMR90 or LF1 cells were transfected using Lipofectamine RNAimax (Life technologies) (CAT#13778150) with 200 pmol of either ON-TARGET plus Non-targeting pool (D-001810) or with a pool of siRNA sequences purchased from Sigma-Aldrich targeting *UBE2E1* (pool of 2 oligonucleotides, SASI_Hs01_00147125, SASI_Hs01_00335491) *UBE2E2* (pool of 2 oligonucleotides, SASI_Hs01_00102040, SASI_Hs01_00102041), *UBE2E3* (pool of 2 oligonucleotides, SASI_Hs01_00107204, SASI_Hs01_00107206), *ASXL2* (pool of 3 oligonucleotides SASI_Hs01_00202197, SASI_Hs01_00202200, SASI_Hs01_00202201), *PSMD14* (pool of 2 oligonucleotides, SASI_Hs01_00024447, SASI_Hs01_00024446), *UCH37* (pool of 2 oligonucleotides, SASI_Hs01_00142742, SASI_Hs01_00142743). ON-TARGETplus SMARTpool *UBE2O* siRNA (pool of 4 oligonucleotides, J-008979-08, J-008979-07, J-008979-06, J-008979-05) was purchased from Dharmacon (L-008979-00-0050). Four days post-transfection, cells were harvested for immunoblotting or three days post-transfection, cells were plated for survival assay (MTT) or for colony forming assay (CFA). U-2 OS, IMR90, LF1 or 3T3L1 cells were transduced with lentiviral expression vectors for HA-ASXL2 WT, HA-ASXL2 K370R, Myc-ASXL2 WT, Myc-ASXL2 K370R, GFP-Myc, GFP-DEUBAD (ASXL2) WT, and GFP-DEUBAD (ASXL2) K370R. 48 h post-selection (2 μg/ml puromycin or 12 μg/ml blasticidin), the cells were plated either for MTT or CFA assays, or plated for cell cycle analysis by synchronizing the cells by double thymidine block (2 mM for each block) or nocodazole (200 ng/ml). Effectene transfection kit (Qiagen, CAT#301427) was used to conduct ectopic expression experiments in S2 cells[57]. Briefly, $3 \times 10^6$ S2 cells were plated in 6 cm dishes and incubated overnight at 25 °C. A total of 1 μg of plasmid DNA was then transfected according to manufacturer's instructions.

**Colony forming assay (CFA) and MTT assay**. Similar numbers of U-2 OS or 3T3L1 ($50 \times 10^3$) cells stably expressing the different constructs of ASXL2 were seeded on the plates and cultured for 5 to 10 days. The surviving colonies were washed with PBS and fixed with 3% paraformaldehyde (PFA) for 20 min. Cells were then washed once with PBS and stained with 0.2% crystal violet for 10 min followed by several washes with water. MTT (3-(4,5-Dimethylthiazol-2-yl)-2,5-diphenyltetrazolium bromide) viability assay was conducted as follows[58]. Briefly, Cells were transduced with various lentiviral constructs as denoted in the figure legends. At the indicated time points post-puromycin selection, the medium was removed and replaced with new medium containing 200 μg/ml of MTT and the cells were then incubated at 37 °C for 2 h. The purple colored formazan product was extracted with DMSO at the end of the incubation. The absorbance was then measured at 490 nm using a microplate reader (Biotek Instruments), and the results were expressed as the percentage of MTT conversion relative to the absorbance of the control cells.

**Fly Stocks, transgenesis, and microscopy**. The *Drosophila* strains, w[1118]; P{w[+ mW.hs] = en2.4-GAL4}e16E, P{w[+ mC] = UAS-2xEGFP}AH2 (BL-25752) and w[*]; P{w[+ m*] = Ubi-GAL4.U}2/CyO (BL-32551) were obtained from the Bloomington Stock center. Crosses were performed at 25 °C on standard fly food (Jazz-Mix, Fisher Scientific). Transgenic flies were generated by targeted PhiC31-mediated germline transformation[59]. Two *Drosophila* lines with an AttP landing site at position 75A10 (BL-24862) and 65B2 (BL-24871) were used for germline transformation.

Images of flies and halteres were generated using a Leica stereomicroscope equipped with a Canon camera. Images of wings were generated with a Nanozoomer (Hamamatsu). For immunohistochemistry of larval discs, larvae were dissected in Schneider's insect medium supplemented with 1 mM CaCl₂, fixed for 15 min in 4% PFA in PBS supplemented with 1 mM CaCl₂ at room temperature and washed three times in PBS, TritonX-100, 0.2 % (PBS-T). The dilutions for primary antibodies are as follow: H2Aub (1/200), UBX (1/200), Flag M2 (1/200), cleaved Caspase-3 (1/200). Discs were incubated with primary antibodies diluted in

PBS-T containing 5 % BSA overnight at 4 °C. Discs were then washed three times in PBS-T and incubated at room temperature for 2 h with fluorophore-conjugated secondary antibodies (1/500; Molecular Probes) diluted in PBS-T. Discs were then washed once in PBS-T containing 100 ng/ml DAPI, twice in PBS-T, and mounted in Mowiol (Sigma). Imaging was done with a Zeiss LSM700 confocal microscope, equipped with 20× (Fig. 8a and Supplementary Fig. 8a) or 40× (Supplementary Fig. 8c) objectives.

**dsRNA production and *Drosophila* protein analysis**. RNAi in S2 cells was conducted according to standard procedures[60]. Briefly, T7 RNA polymerase was used for in vitro transcription of all dsRNA used in this study. Following Sodium acetate (NaOAc)/ethanol precipitation, dsRNA concentration was assessed by Nanodrop. Individual dsRNA were added to cells at a final concentration of 10 µg/ml and left for 4 days on cells before harvesting. Cells were lysed 72 h after transfection in (25 mM Tris pH 7.3 and 2 % sodium dodecyl sulfate (SDS)) buffer and western blots were performed. For evaluation of expression levels of Asx WT or Asx K325R in whole larva, 10 larvae per genotype were crushed in 100 µl of RIPA buffer. Lysates were centrifuged and protein content in the supernatant was evaluated. 20 µg of protein were used for western blot analysis.

**siRNA screen**. Human DUBs siRNA library (Supplementary Table 1) was purchased from Sigma (two siRNA oligos were pooled for each DUB). U-2 OS cells, stably expressing pOZ-Flag-HA-BAP1 and pLenti-CMV-GFP-DEUBAD (ASXL2), were transfected with individual siRNA pool targeting DUBs using Lipofectamine RNAiMax. The cells were transfected twice within 24 h. Four days post-transfection, cells were harvested for immunoblotting. For the siRNA E2 screen, 42 siRNA pools were purchased from Sigma (three siRNA oligos for each E2) (Supplementary Table 2). U-2 OS cells stably expressing pLenti-CMV-GFP-DEUBAD (ASXL2) were transfected with individual siRNA pools two times. Four days later, cells were harvested for immunoblotting.

**CRISPR/Cas9 knockout**. pLentiCRISPR_V2 vector was used to generate *UBE2E3* knock-out cells[61]. We also used a pLentiCRISPR_V2 vector targeting GFP as a control. Briefly, 21 bp of custom synthesized gRNAs oligos with proper overhang ends (Supplementary Table 4) were annealed and ligated into pLentiCRISPR_V2 plasmid previously digested with BsmBI (NEB, #R0580S) restriction enzyme. Positive clones were sequence verified and used to generate lentivirus particles in HEK293T cells by transfection with packaging vectors, psPAX2 and pMD2-G. Media enriched with lentiviral particles was then used to transduce cells at least twice followed by 48 h of puromycin selection. Knockout efficiency was assessed by western blotting. The transduced cells were used for CFA, cell counts and FACS analysis.

**Immunoblotting and antibodies**. Total cell extracts were used for SDS-PAGE and immunoblotting was done according to standard procedures[10]. Briefly, total cell extracts were prepared by lysing cells with buffer containing 25 mM Tris pH 7.3 and 1% sodium dodecyl sulfate (SDS). Cell extracts were boiled at 95 °C for 10 min and then sonicated. Quantification of total proteins was conducted using the bicinchoninic acid (BCA) assay, and samples were diluted in Laemmli buffer. The band signals were acquired with a LAS-4000 LCD camera coupled to MultiGauge software (Fuji, Stamford, CT, USA). See Supplementary Table 5 for a complete list of the antibodies. Please see the Electronic Supplementary Material that accompanies the manuscript for all the uncropped scans for all the blots (Supplementary Fig. 12 to Supplementary Fig. 33).

**Immunoprecipitation**. Immunoprecipitation experiments were done following overexpression in HEK293T cells[10]. Cells were resuspended in 50 mM Tris, pH 7.3; 150 mM NaCl; 5 mM EDTA; 10 mM NaF; 1% Triton X-100; 1 mM PMSF, protease inhibitors cocktail (Sigma-Aldrich), and the lysates were clarified by centrifugation at 21,000 g for 30 min. The supernatants were incubated with 2 µg of the indicated antibodies overnight at 4 °C. The immune complexes were recovered the next day with protein G sepharose beads saturated with 1% BSA, subjected to extensive washes and then resuspended in Laemmeli buffer for western blot analysis.

**In vitro ubiquitination assays**. His-BAP1, His-BAP1 C91S, MBP-DEUBAD (ASXL2) (WT and K370R) and His-tagged UBE2E2 and UBE2E3 proteins were purified from bacteria as described above. For His-BAP1/MBP-DEUBAD (ASXL2) complexes, recombinant His-BAP1 cell lysate was equally mixed with recombinant MBP-DEUBAD (ASXL2) cell lysate. After nickel agarose beads pulldown, the His-BAP1/MBP-DEUBAD (ASXL2) complex was eluted in 50 mM Tris pH 7.3; 500 mM NaCl; 0.2 % Triton; 5 mM DTT; 1 mM EDTA; 1 mM PMSF; protease inhibitors cocktails and 200 mM Imidazole. The eluates were then diluted in 50 mM Tris pH 7.3; 1 % Triton; 5 mM DTT; 1 mM EDTA; 1 mM PMSF and protease inhibitors cocktails to have 300 mM NaCl in the buffer prior to incubation with amylose agarose beads. After washes with the same buffer, the His-BAP1/MBP-DEUBAD (ASXL2) complex was kept on the amylose agarose beads in 50 mM HEPES pH 8.0; 50 mM NaCl; 10 % Glycerol and 1 mM DTT. The in vitro ubiquitination screen assay on the MBP-beads immobilized BAP1/DEUBAD(ASXL2) complex was performed using E1 (250 ng), Ub (50 ng/µl), and 32 E2 Ub-

conjugating enzymes (1 µM) and 2–3 µg of BAP1/DEUBAD(ASXL2) complex. The reactions were incubated for 3 h at 37 °C in 25 mM Tris, pH 7.5; 10 mM MgCl2 and 5 mM ATP. The reactions were stopped by adding Laemmli buffer and analyzed by western blotting. The different His-BAP1 mutant forms (C91S, and the Ub-binding deficient mutants) in complex with the MBP-DEUBAD (ASXL2) WT were purified and used for the in vitro ubiquitination assays as described for His-BAP1 WT/MBP-DEUBAD (ASXL2) WT complex. Labeling of BAP1 and its mutants with HA-Ub-VME probe was conducted using total cell extracts which were prepared in 50 mM Tris pH 7.3, 5 mM MgCl2, 1 mM DTT and 2 mM ATP. About 20 µg of protein extract was used for labeling with Ubiquitin-Vinyl Methyl Ester for 2 h at room temperature. Reactions were stopped by adding Laemmli buffer and analyzed by western blotting.

**Purification of DEUBAD/BAP1 complexes**. HEK293T cells were transfected with either 3xFlag-DEUBAD (ASXL2) WT or 3xFlag-2HA-DEUBAD (ASXL2) K370R, along with Myc-BAP1 and with or without HA-Ub (for DEUBAD (ASXL2) WT and DEUBAD (ASXL2) K370R respectively) expression vectors. Four days later, the cells were harvested for immunopurification. The purification was done as previously described[14]. Briefly, total cell extracts obtained after resuspending cells in 50 mM Tris pH 7.3; 150 mM NaCl; 1 % Triton; 10 mM β-glycerophosphate; 1 mM Na3VO4; 20 mM NEM; 1 mM EDTA; 1 mM PMSF and protease inhibitors cocktail, were used for the immunopurification. The extracts were first clarified by centrifugation at 30,000 g for 1 h, followed by filtration through a 0.45 µm pore filter. The lysates were then incubated with the anti-Flag M2 resin (CAT#A2220, Sigma Millipore) overnight and extensively washed with the lysis buffer. The resin was eluted three times with the same buffer containing 200 ng/ml of Flag peptide (CAT#F3290, Sigma). The eluted fractions were incubated with anti-HA resin (CAT#A2095, Sigma Millipore) overnight, and the procedure was repeated as for the previous column. Native monoubiquitinated-H2A nucleosomes were purified as previously described[33]. Following transfection of HEK293T with pCDNA.3 Flag-H2A, chromatin was fractionated following incubation of cell in 50 mM Tris·HCl, pH 7.3, 420 mM NaCl, 1% Nonidet P-40, 1 mM PMSF, protease inhibitor mixture; Sigma and 20 mM N-ethylmaleimide (NEM). The chromatin fraction was washed three times with the same buffer followed by two washes using MNase buffer (20 mM Tris·HCl, pH 7.3, 100 mM KCl, 2 mM MgCl2, 1 mM CaCl2, 0.3 M sucrose, 0.1% Nonidet P-40, 1 mM PMSF, protease inhibitor mixture). Soluble nucleosomes were then obtained following digestion with with 3 U/ml of micrococcal nuclease (MNase) for 10 min at room temperature. The reaction was stopped with 5 mM EDTA, and the soluble chromatin fraction was incubated overnight at 4 °C with Flag M2 agarose beads (Sigma). Beads were then washed several times with 50 mM Tris·HCl, pH 7.3, 5 mM EDTA, 300 mM NaCl, 10 mM NaF, 1% Nonidet P-40, 1 mM PMSF, 1 mM DTT, and protease inhibitors mixture (Sigma) containing 20 mM NEM followed by several washes with the same buffer without NEM. Bead-bound nucleosomes were then eluted with Flag peptides (0.2 µg/ml).

**In vitro deubiquitination assays**. The deubiquitination reactions were done using monoubiquitinated-Flag-DEUBAD (ASXL2) WT-HA-Ub or Flag-HA-DEUBAD (ASXL2) K370R each in complex with BAP1. These complexes were incubated with monoubiquitinated-H2A nucleosomes in 50 mM Tris pH 7.3; 1 mM MgCl2; 50 mM NaCl, and 1 mM DTT. The DUB reactions were stopped at the indicated times by adding Laemmli buffer and analyzed by western blotting.

**Stable cell lines expressing DEUBAD, BAP1, or ASXL2**. U-2 OS cell line stably expressing HA-ASXL2 WT or HA-ASXL2 K370R were generated by lentiviral infection using respective pLenti-CMV constructs followed by selection by 2 µg/ml of puromycin. U-2 OS cell lines stably expressing GFP-DEUBAD (ASXL2) WT, GFP-DEUBAD (ASXL2) K370R, Flag-HA-BAP1-GFP-DEUBAD (ASXL2) WT, Flag-HA-BAP1-GFP-DEUBAD (ASXL2) K370R were generated following lentiviral infection using pLenti-CMV constructs expressing respectively GFP-DEUBAD (ASXL2) WT and GFP-DEUBAD (ASXL2) K370R and selection with 12 µg/ml of Blasticidin. U-2 OS, IMR90, LF1 and 3T3L1 cell lines stably expressing Myc-GFP, Myc-ASXL2 WT and Myc-ASXL2 K370R were generated by transduction using lentiviral pLenti-CMV expression constructs. The cells were selected using puromycin (2 µg/ml for U-2 OS, 3 µg/ml for IMR90 and LF1 and 4 µg/ml for 3T3L1).

**Synchronization and cell cycle analysis**. U-2 OS cells were synchronized at the G1/S using double thymidine block method[62]. Briefly, cells were treated with 2 mM thymidine for 12 h followed by 12 h release in fresh medium and then treated with 2 mM thymidine for 12 h. Cells were then released into new media and analyzed for cell cycle, at the indicated time points, to follow S phase progression. To study the progression of the cells through the cell cycle, cells were treated with 200 ng/ml of Nocodazole and followed by cell cycle analysis at different times point[62]. 3T3L1 cells were treated with nocodazole (200 ng/ml) for the indicated times and harvested for cell cycle analysis. For FACS analysis[62], the cells were washed with PBS and then harvested by trypsinization and fixed with 70 % ethanol. Following centrifugation and resuspension in PBS, cells were treated with 100 µg/ml RNase A for 30 min at 37 °C and stained with 50 µg/ml propidium iodide. DNA content of cells was acquired with BD (Becton Dickinson) FACS Calibur: 2 lasers, 4 detectors

and analyzed using a FACScan flow cytometer fitted with CellQuestPro software (BD Biosciences). The unprocessed FACS profiles (Supplementary Fig. 34 to Supplementary Fig. 37) and an exemplifying figure for cell population gating (Supplementary Fig. 38) are provided in the Electronic Supplementary Figures.

**Immunofluorescence.** The immunofluorescence experiments were performed using the indicated antibodies and according to standard procedures[63]. Briefly, cells were fixed in 3% PFA-PBS for 20 min and permeabilized using PBS with 0.5% NP-40. Following blocking with PBS with 0.1% NP-40 and containing 10% FBS, the coverslips were incubated with the antibodies as indicated for two to three hours (anti-H2Aub at 1/5000 and anti-Myc antibody at 1/1000). Anti-mouse Alexa Fluor® 594, anti-mouse Alexa Fluor® 488, Anti-rabbit Alexa Fluor® 488, or anti-rabbit Alexa Fluor® 594 (Life Technologies) were used as secondary antibodies at 1/2000 and nuclei were stained with 4′,6-diamidino-2-phenylindole (DAPI). Images were acquired using BX53 OLYMPUS microscope U-HGLGPS, XM10 digital monochrome camera and UPlan SApo 60 × /1,35 Oil objective. Images were processed using WCIF-ImageJ software (NIH).

**Identification of ASXL2 ubiquitination sites.** HEK293T ($\sim 1 \times 10^9$) cells were transfected with Flag-ASXL2 or Flag-ASXL1 expression vectors. Four days later, the cells were harvested for immunopurification. Total cell extracts were obtained by resuspending cells in 50 mM Tris pH 7,3; 300 mM NaCl; 1 % Triton; 10 mM β-glycerophosphate; 1 mM Na3VO4; 20 mM NEM; 1 mM DTT; 1 mM EDTA; 1 mM PMSF and protease inhibitors cocktail. The immunopurified complexes were obtained as described above (in the in vitro deubiquitination assay section). The eluted fractions were combined and concentrated using trichloroacetic acid (TCA) precipitation. The purified proteins were loaded for Coomassie gel staining and the bands corresponding to ASXL1 and ASXL2 were excised and analyzed by MS. Bands were de-stained in acetonitrile (ACN) 50%. Samples were diluted with 100 mM ammonium bicarbonate and 5 mM tris (2-carboxyethyl)phosphine (TCEP) and vortexed at 37 °C for 30 min. A solution of 110 mM chloroacetamide in 100 mM ammonium bicarbonate was added to get a final concentration chloracetamide of 55 mM. Samples were vortexed for 30 min at 37 °C. Trypsin was added to reach an enzyme to protein ratio of 1/50 and the digestion was performed overnight. Peptide extraction was performed with 90% ACN. Samples were then dried down in a Speed-Vac and reconstituted in 40 μL of formic acid 0.2%. Tryptic peptides were loaded on a C18 stem trap (New Objective) and separated on a home-made C18 column (15 cm × 150 μm id) at a flow rate of 600 nL/min with a gradient of 5–30% B (A: formic acid 0.2% in water, B: formic acid 0.2% in ACN). The analytical column was coupled to a Q-Exactive Plus (Thermo Fisher Scientific). The resolution was set at 70,000 for the survey scan and 17,500 for the tandem mass spectrum acquisition. A maximum of 12 precursors were sequenced for each duty cycle. The automatic gain control (AGC) target values for MS and MS/MS scans were set to 3e6 (max fill time 50 ms) and 2e4 (max fill time 150 ms), respectively. The precursor isolation window was set to m/z 1.6 with a high-energy dissociation (HCD) normalized collision energy of 25. The dynamic exclusion window was set to 30 s. Tandem mass spectra were searched against the UniProt human database with carbamidomethylation (C) as fixed modifications, deamidation (NQ,) oxidation (M) acetylation (N-term) and ubiquitination (GG-tagged Lysines) as variable modifications. Tolerance was set at 10 ppm on precursor mass and 0.01 Da on fragment ions. All MS spectra figures were generated using Scaffold 4.8.6 software (Proteome Software, Portland, OR, USA).

**Protein sequence analysis and structure modeling.** Sequence alignments of multiple ASXLs orthologs as well as UBE2Es sequences were performed using Aline[64]. A homology model of BAP1 in complex with Ub and the DEUBAD domain of ASXL2 was generated from the crystal structure of UCH37 in complex with Ub and DEUBAD domain of RPN13 (PDB:4UEL)[32,65]. The DEUBAD (ASXL2) chain was modeled by manually substituting RPN13 amino acids according to a structure-based sequence alignment using Coot[66]. A BAP1 model lacking a large loop region and the NLS was generated by SWISS Model[67], and this was followed by substitution of amino acid differences superimposed on the UCH37 structure using Coot. Geometry minimization of the final homology model complex was performed using a module within Phenix[68]. Structural figure showing the BAP1/Ub/DEUBAD (ASXL2) homology model were generated by PyMol (Schrodinger, LLC. 2010. The PyMOL Molecular Graphics System, Version 1.8.0.5).

**Human mesothelioma immunohistochemistry.** Collection and use of patient samples were approved by the study protocols no. CHS14406 (University of Hawaii Institutional Review Board) and no. i8896 (New York University Institutional Review Board). Written consent was received from all patients. A characterization of human participants (age, gender, genotypic information, and therapeutic treatment) is provided as a Supplementary Table 7. Formalin-fixed paraffin embedded tissue sections were stained using the avidin-biotin-peroxidase complex method in a DAKO-autostainer (Carpinteria, CA, USA). The primary antibodies used were: BAP1 (1:50), UBE2E1/UbcH6 (1:600), UBE2E2 (1:200), UBE2E3 (1:150), ASXL2 (1:250) (see Supplementary Table 5 for a complete list of the antibodies). Diagnoses were made on hematoxylin-eosin stained sections combined with immunohistochemical features.

**Statistics.** Most of the experiments were repeated at least three time unless otherwise is stated. DEUBAD (ASXL2), BAP1 and H2Aub bands quantifications were done using ImageJ software. Obtained values were normalized to the siNT and presented as indicated. For Fig. 4b, log2 transformed values was performed using a custom-made R script and Heatmaps were generated using the Complex Heatmaps package[69]. DUBs and E2s RNAi screen was conducted once. For patient tumor staining, we used logistic regression to calculate associations between continuous variables and the dichotomous BAP1 variable[70]. Between two continuous variables, we calculated Pearson correlations.

## Data availability

The mass spectrometry proteomics data have been deposited to the ProteomeXchange Consortium via the PRIDE partner repository with the dataset identifier PXD011124 and 10.6019/PXD011124. The other data that support the findings of this study are available within the article or its supplementary information and from the corresponding author upon reasonable request.

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

## Acknowledgements

We thank Jürg Muller for insightful comments and reagents. We thank Diana Adjaoud, Ian Hammond-Martel, Pham My-An, and Erlinda Diaz Fernandez for technical assistance. This work was supported by grants from the Canadian Institutes of Health Research (CIHR) to E.B.A. (399244), a Foundation grant from the CIHR to M.T. (388023), a Foundation grant from the CIHR to J.Y.M. (388879), and grants from the Natural Sciences and Engineering Research Council of Canada (2015-2020) and Mesothelioma Applied Research Foundation (MARF) to E.B.A. E.B.A. is a senior scholar of the Fonds de la Recherche du Québec-Santé (FRQ-S). J.Y.M. is a Fonds de la Recherche du Québec-Santé (FRQ-S) research chair in genome stability . This work was supported by Department of Defense Grant No. CA150220 toH.Y. and M.C.; National Cancer Institute (NCI) GrantNo. R01 CA198138 to M.C.; the University of Hawaii Foundation, whichreceived unrestricted donations to support cancer and mesotheliomaresearch from: The Melohn family endowment (M.C.); HoneywellInternational (M.C.); The Riviera United 4-a Cure to M.C. and H.Y. S.D. has a Banting postdoctoral fellowship. H.B. has a PhD scholarship from the Ministry of Higher Education and Scientific Research of Tunisia and the Cole Foundation. O.A. has a MSc scholarship from the The Canadian Francophonie Scholarship Program. The Institute for Research in Immunology and Cancer (IRIC) receives infrastructure support from Genome Canada and Génome Québec, IRICoR, the Canadian Foundation for Innovation, and the Fonds de Recherche du Québec - Santé (FRQS).

## Author contributions

S.D., H.B., M.T., and E.B.A. designed the experiments. S.D., H.B., O.A., L.M., N.S.N., M.U., D.T., and N.M. performed and analyzed most of the experiments on mammalian cells under the supervision of E.B.A. H.B. generated the transgenic flies, H.B. and C.B. designed, performed, and analyzed all experiments with *Drosophila* under supervision of M.T. and E.B.A. E.B. performed the MS analysis for ASXL1/2 complexes under the supervision of P.T., S.D., and D.C. performed the structural analysis under supervision of F.S. H.Y. and M.T. designed and performed the mesothelioma immunohistochemistry (IHC) experiments and H.Y. and M.C. interpreted the ICH data. M.U. conducted part of the experiments on ubiquitination in vitro under supervision of J.Y.M. and E.B.A. All authors participated conceptual discussions. S.D., H.B., and E.B.A. compiled the figures and wrote the manuscript.

## Additional information

**Competing interests:** The authors declare no competing interests.

