## [Peer Review File · Nature Communications]

Reviewers' Comments:

Reviewer #1:

Remarks to the Author:

This is an impressive manuscript that studies the mono-ubiquitination of the DEUBAD domain of ASXL2 (or Asx in *Drosophila*) at a specific site (K370). They show that ubiquitination in the free protein leads to its polyubiquitination and clearance by the proteasome, but that BAP1 stabilizes the mono ubiquitinated form of ASXL2 and this does not get polyubiquitinated. They identify the two E2 enzyme that makes this ubiquitination and they show that the ubiquitination of ASXL2 activates the DUB activate of the BAP1, in vitro and in cells. They then show that this modification is relevant, but not absolutely critical for the developmental roles of the BAP1/Asx complex in *Drosophila*, indicating some kind of treshold effect, but that it has strong effects on the mammalian cell cycle phenotype. Finally they show a correlation with mesothelioma. Altogether this is a real tour-de-force and the data are consistent and convincing. Mechanistically this type of ubiquitin regulated DUB activity is very interesting. The relevance of the site is very strongly underpinned by the unbiased data in Phosphosite, that confirm the relevance of the K370 ubiquitination. The consequences of the modification including the partial phenotype in *Drosophila* and the difference with mammals fits with the difference in higher order complexes between *drosophila* and mammals, and reconfirms the complexity of BAP1 phenotype. Given the importance of BAP1 loss in a series of different human tumors understanding its regulation is important. This paper certainly deserves rapid publication in a major journal.

There are a few minor issues that could be discussed for further clarification and/or improved in the manuscript;

- The fact that in vitro no E3 ligase is needed is not proof that there is none in cells. In fact this is reminiscent of the situation for SUMO, where identification of E3s lagged behind, but eventually were shown to be critical even for most SUMO consensus motifs. It's good to clarify this in the text
- The mapping of the site and the DEUBAD domain are not consistent throughout the paper. There seems to be a historical fragment where the UCH37/RPN13 structural knowledge is not used, and it would help to rewrite this section
- It's not really correct to refer to ub-VME 'binding' as this is a suicide substrate that involves enzyme activity and doesn't release.
- The levels of ubiquitination of the GFP-fusion are lower than expected, even in the presence of BAP1. This could be discussed a bit more extensively, including the possibility of the GFP interfering with modification.
- It could be helpful to refer to the Phosphositeplus independent proteomics data where this modification is clearly very prevalent.
- Fig 7A-c: it would be good to have controls for levels of BAP1 expression
- Fig 6e: surely the y-axis must be in % ? please correct the legend
- Sup fig 1: Some of the spectra have lost their x-axis.
- Sup figure sf1d and e should be formatted like sf1a.
- On page 17 after the reference to Fig. 6a, deplete -> depletes
- Page 25 DEUBD -> DEUBAD
- Page 35 similar numbers -> state how many
- Page 35 5 to10 -> 5 to 10
- Page 43 What software was used for mass spec peak conversion to peptide / protein lists?
- It is now unclear whether the search was for Kg (as stated) or Kgg (as is normal) residues.
- For figure 1c please add size indication (kDa) to the figure.

Reviewer #2:

Remarks to the Author:

The results of Dauo et al. convincingly show that Asx1s and Asx are mono-ubiquitinated and that this helps control the deubiquitinase activity of Bap1/Calypso. Their results are succinctly summarized in the model shown in Fig. 10. I found the introduction and results very logical and easy to follow, and, in general, the results convincing. Very nice work! The discussion was very repetitive with the results and could be substantially shortened.

Regarding the Drosophila results:

The authors have shown that overexpression of Asx by en-Gal4 causes a loss of H2Aub, and a loss of Ubx in the HD in a patchy manner (see Fig. 8), consistent with the phenotype seen. In contrast, overexpression of Asx K325R causes an apparent increase in H2Aub, and a dramatic loss of Ubx expression. It should also be noted that expression of the en-Gal4 gene is also disrupted by Asx K325R mutation; GFP expression in en-Gal4, U-GFP Asx K325R discs is patchy. Especially evident in Fig. S7. The Actin control is saturated in Fig. S7, so hard to directly compare expression levels, but close enough. No way to compare with endogenous Asx levels, but I am guessing it is overexpression.

The Drosophila results support the conclusion that Asx K325R acts in a dominant negative manner and that altering H2Aub levels disrupts Ubx expression. However, because these experiments involve overexpression, one cannot conclude, as stated in the abstract "...monoubiquitination ...is required for Drosophila development." This has not been shown. What they have shown is that over-expression of either Asx or Asx K325R disrupts development. Asx K325R overexpression causes more severe effects, but Asx overexpression also causes developmental defects, consistent with the interpretation by others that it is the turnover of H2Aub that may be important for regulating gene expression, not the absolute level of H2Aub per se. To show this K325 residue is required for Drosophila development, they would need to do a rescue experiment with the wildtype and K325R mutant Asx proteins. I suggest just changing the statement in the abstract to accurately represent the results.

Reviewer #4:

Remarks to the Author:

General comments

The authors present a model to describe the activation mechanism of BAP1 allowing deubiquitination (DUB catalysis) They propose that the conserved DEUBAD domain of the ASXL2 is monoubiquitinated by UBE2Es (3), stabilizing it, in order to activate BAP1's DUB activity required for H2A deubiquitination and polycomb gene repression.

Overall, this manuscript presents a comprehensive series of well designed and executed experiments that lend support to this model, including evidence of evolutionary conservation of the ASX DEUBAD domain mono ubiquitination by the BAP1 ortholog Calypso in Drosophila.

The paper provides strong evidence to underpin its conclusions. The data is novel and provides an explanation for BAP1 activation which is significant. For scientists in this field. The results are of importance. Because of the implications for BAP1 inactivation across a range of cancers, this data will be of interest to researchers in other related disciplines. I support the publication of this manuscript with some suggested amendments.

Specific Comments

The functional impact of ASXL2 monoubiquitination is studied and the authors show that K370 mutant or depletion results in reduced cellular proliferation/colony formation, presumably by blocking BAP1 DUB activation. This effect appears to be phenocopied in UBE2E3 deficient cells. Presumably, by extension, given that these interventions should phenocopy BAP1 inactivation, this should be confirmed in this model system (U-2 OS cells) eg. BAP1 depletion phenocopies the

reduction in proliferation as this is important missing data.

In examining mesothelioma, no hypothesis is stated for the correlative experiments. This results section is the least clear in the paper and the implications for these studies could be elaborated. For example, was there quantitative evidence of ASXL2 destabilization in BAP1 negative/mutant cells. For example, did pathogenic mutants of BAP1 capable of disrupting AXSL1-UBE2E3 axis that correlated with ASXL2 expression? The logistic regression analysis although significant needs some clarification in terms of interpretation.

The evidence that pathological BAP1 CTD mutant BAP1 Δ R666-H669 is associated with loss of DEUBAD ubiquitination as well as loss of interaction is of interest. More than half of pathogenic BAP1 mutations map to UCH It would be good to show (perhaps in a graphic) the frequencies of cosmic and germ line single nucleotide variants in relation to CTD and CC1 in BAP1, with potential to disrupt endogenous DEUBAP monoubiquitination. Does the mutation spectra implicate disruption of the DEUBAD-ub ASXL2/UBE3E3 axis universally in cancers such as mesothelioma?

There is phylogenetic conservation of protein partners in the other UCH family protein ie. UCH-L5 which is associated with RPN13 and INO80G which harbour DEUBAD domains. It would be of interest if these are similarly monoubiquitinated as a common mechanism of DUB activation. This would be expected given the highly conserved monoubiquitination of Drosophila ASX.

It would be of interest, given the extensive genomic data available publically, to comment on whether or not DEUBAD domain/ASXL2 or UBE2E2 mutations are seen in cancer and whether there is mutually exclusivity with BAP1 SNVs.

Response to Reviewer's Comments

We sincerely thank the reviewers for their time, enthusiasm and very insightful comments. **Reviewer 1 stated:** "This is an impressive manuscript that studies the mono-ubiquitination of the DEUBAD domain of ASXL2 (or Asx in Drosophila) at a specific site (K370). Altogether this is a real tour-de-force and the data are consistent and convincing. Mechanistically this type of ubiquitin regulated DUB activity is very interesting. Given the importance of BAP1 loss in a series of different human tumors understanding its regulation is important. This paper certainly deserves rapid publication in a major journal". **Reviewer 2 stated:** "The results of Dauo et al. convincingly show that Asx1s and Asx are mono-ubiquitinated and that this helps control the deubiquitinase activity of Bap1/Calypso. Their results are succinctly summarized in the model shown in Fig. 10. I found the introduction and results very logical and easy to follow, and, in general, the results convincing. Very nice work!". **Reviewer 4 stated:** "Overall, this manuscript presents a comprehensive series of well-designed and executed experiments that lend support to this model. The paper provides strong evidence to underpin its conclusions. The data is novel and provides an explanation for BAP1 activation which is significant. For scientists in this field. The results are of importance. I support the publication of this manuscript with some suggested amendments".

We are delighted that the editor invited us to resubmit a revised manuscript. Indeed, despite the overall appreciation, the reviewers raised some important concerns that we have now addressed.

Reviewer 1

Comment 1: "The fact that in vitro no E3 ligase is needed is not proof that there is none in cells. In fact this is reminiscent of the situation for SUMO, where identification of E3s lagged behind, but eventually were shown to be critical even for most SUMO consensus motifs. It's good to clarify this in the text".

Response 1: We thank the reviewer for this important point, and we agree that we cannot completely exclude that an ubiquitin E3 ligase is still needed to direct the ubiquitination of ASXLs in vivo. Further discussion about the role of additional E3 ligases in regulating ASXL2 DEUBAD mono- and poly-ubiquitination has now been added in the text (section discussion). We also appreciated the reviewer comment on the parallel that we can make between UBE2E-mediated ubiquitination and Ubc9-mediated SUMOylation. Thus, we also included a small description in the discussion about this interesting point.

Comment 2: "The mapping of the site and the DEUBAD domain are not consistent throughout the paper. There seems to be a historical fragment where the UCH37/RPN13 structural knowledge is not used, and it would help to rewrite this section".

Response 2: We now have more carefully homogenized the paper and included in the introduction (see introduction), the historical description of the DEUBAD domain in respect to UCH37(UCHL-5)/INO80G/RPN13 paradigm. Since there was also a comment of **reviewer 4** regarding whether the monoubiquitination of ASXL2 DEUBAD by BAP1 could be a general mechanism of activation adopted by other UCH DUB members, we conducted additional experiments to address this point. We investigated whether UCH37 could promote monoubiquitination of both RPN13 and INO80G DEUBAD domains. As presented in the **Supplementary Figure 2e, f**, and stated in the results section, our data suggest that UCH37 does not promote ubiquitination of RPN13 or INO80G DEUBAD domains. Thus, it seems that the effect of DUB on DEUBAD monoubiquitination is a specific mechanism of BAP1/ASXLs complexes.

Comment 3: “It’s not really correct to refer to ub-VME ‘binding’ as this is a suicide substrate that involves enzyme activity and doesn’t release”.

Response 3: We believe that, in general, proper ubiquitin binding (involving key interfaces of BAP1 or Ubiquitin), precedes and is required for Ub-VME probe to be covalently linked to the catalytic site. Nonetheless, this has now been revised in the text to avoid confusion (see results section).

Comment 4: “The levels of ubiquitination of the GFP-fusion are lower than expected, even in the presence of BAP1. This could be discussed a bit more extensively, including the possibility of the GFP interfering with modification”.

Response 4: We agree with the reviewer. In fact, the levels of ASXL2 monoubiquitination with HA-Ub are always higher than those with GFP-Ub. The low level of GFP-Ub conjugation could be due to several possibilities that we did not evaluate, as outside the scope of this study. Nonetheless, we have now briefly discussed that the GFP tag might interfere with ubiquitin charging/ligation. On the other hand, the cellular levels of GFP-Ub might not be optimal as that of HA-Ub or untagged ubiquitin (see results section).

Comment 5: “It could be helpful to refer to the PhosphoSitePlus independent proteomics data where this modification is clearly very prevalent”.

Response 5: We believe that we already highlighted that ASXL2 K370 ubiquitination was previously identified in previous global proteomic studies. Nonetheless, we have now better emphasized this in **supplementary Figure 1b** based on the *PhosphoSitePlus* database. This now provides a more precise idea about the prevalence of DEUBAD monoubiquitination.

Comment 6: “Fig 7A-c: it would be good to have controls for levels of BAP1 expression”.

Response 6: This is an important point. We now added immunostaining of BAP1 as a **Supplementary Figure 6** for the same experiments presented in **Figure 7a-c**. The corresponding results description are included in the results section.

Comment 7: “Fig 6e: surely the y-axis must be in % ? please correct the legend”.

Response 7: This has now been corrected in the figure.

Comment 8: “Sup fig 1: Some of the spectra have lost their x-axis. Sup figure sf1d and e should be formatted like sf1a”.

Response 8: We have now corrected and homogenized all the MS spectra in the figures (see **supplementary Figure 1a,d**).

Comment 9: “On page 17 after the reference to Fig. 6a, deplete -> depletes. Page 25 DEUBD -> DEUBAD. Page 35 similar numbers -> state how many. Page 35 5 to10 -> 5 to 10”.

Response 9: These have been corrected as suggested.

Comment 10: “Page 43 What software was used for mass spec peak conversion to peptide / protein lists?”.

Response 10: We have used the Scaffold 4.8.6 proteomics package. This has been added in the material and methods

Comment 11: “It is now unclear whether the search was for Kg (as stated) or Kgg (as is normal) residues”.

Response 11: The search was for Kgg. This has been clarified in the text.

Comment 12: “For figure 1c please add size indication (kDa) to the figure”.

Response 12: We appreciate the reviewer's concern. We believe that adding the molecular weight for the **Figure 1c** will require adding the corresponding size indication (kDa) to all the figures for consistency. As this will render the figures crowded and hard to read, we preferred to add the molecular weight indication to the non-cropped blots for all the figures. Please see the **Electronic supplementary material** that accompanies the manuscript.

Reviewer 2

Comment 1: “The discussion was very repetitive with the results and could be substantially shortened”.

Response 1: We agree with the reviewer. We have made additional changes to remove the redundancy with the results section. We believe that the discussion is now more concise.

Comment 2: “The authors have shown that overexpression of Asx by en-Gal4 causes a loss of H2Aub, and a loss of Ubx in the HD in a patchy manner (see Fig. 8), consistent with the phenotype seen. In contrast, overexpression of Asx K325R causes an apparent increase in H2Aub, and a dramatic loss of Ubx expression. It should also be noted that expression of the en-Gal4 gene is also disrupted by Asx K325R mutation; GFP expression in en-Gal4, U-GFP Asx K325R discs is patchy. Especially evident in Fig. S7. The Actin control is saturated in Fig. S7, so hard to directly compare expression levels, but close enough. No way to compare with endogenous Asx levels, but I am guessing it is overexpression”.

Response 2: We agree with the reviewer and we appreciate these very interesting comments. First, we have conducted a new immunoblotting experiment monitoring actin levels to obtain non-saturated signals. We found that for similar actin levels, Asx WT and Asx K325R proteins are expressed to comparable levels, thus confirming our original conclusion. We have therefore replaced the original panel of actin levels with a less exposed image (**Supplementary Figure 8b**). With respect to the relative levels of transgene expression, using a homemade anti-Asx antibody, we were not able to detect consistently the levels of exogenous versus endogenous Asx. The available antibodies that we used produced substantial background and no distinct Asx band could be recognized (data not shown). Nonetheless, taking into account the expression system used, we believe that Asx K325R might mainly act in a dominant negative manner and we have now changed our statement in the text. On the other hand, we also agree that en-Gal4, U-GFP, U-Asx K325R expression in discs is patchy. While *en-Gal4* irregular activity is readily observable to a weak extent in control or WT Asx-expressing wing discs, it is visibly exacerbated in Asx K325R-expressing discs (**new Supplementary Figure 8c**). The underlying cause is currently unknown, but it does not result from ectopic apoptosis, as no signal for cleaved Caspase-3 could be detected (Eiger expression was used as a positive control for cleaved Caspase-3 signal induced in dying cells) (**new Supplementary Figure 8c**). As suggested by the reviewer, this could also be due to an impact of Asx K325R on the expression of *en-Gal4*. While determining the cause of GFP's patchy expression will require future investigations, we nonetheless clearly observe enhanced H2A-Ub staining specifically in each Asx K325R-expressing cells, thus confirming a cell autonomous effect for this dominant negative variant of Asx. The text has been modified accordingly.

Comment 3: “The *Drosophila* results support the conclusion that Asx K325R acts in a dominant negative manner and that altering H2Aub levels disrupts Ubx expression. However, because these experiments involve overexpression, one cannot conclude, as stated in the abstract “...monoubiquitination ...is required for *Drosophila* development.” This has not been shown. What they have shown is that over-expression of either Asx or Asx K325R disrupts development. Asx K325R overexpression causes more severe effects, but Asx overexpression also causes developmental defects, consistent with the

interpretation by others that it is the turnover of H2Aub that may be important for regulating gene expression, not the absolute level of H2Aub per se. To show this K325 residue is required for Drosophila development, they would need to do a rescue experiment with the wildtype and K325R mutant Asx proteins. I suggest just changing the statement in the abstract to accurately represent the results”.

Response 3: We thank the reviewer for this important comment and we fully agree. The statement has been changed in the abstract as well as in the results and discussion sections. Please also see response immediately above

Reviewer 4

Comment 1: “The functional impact of ASXL2 monoubiquitination is studied and the authors show that K370 mutant or depletion results in reduced cellular proliferation/colony formation, presumably by blocking BAP1 DUB activation. This effect appears to be phenocopied in UBE2E3 deficient cells. Presumably, by extension, given that these interventions should phenocopy BAP1 inactivation, this should be confirmed in this model system (U-2 OS cells) eg. BAP1 depletion phenocopies the reduction in proliferation as this is important missing data”.

Response1: We agree with the comment. Indeed, we previously reported (Daou et al, J Biol Chem. 2015; 290(48):28643-63) that BAP1 depletion affects U-2 OS cell proliferation. Nonetheless, it is always relevant to show, in the same experimental conditions, the appropriate controls. We now present, as a **Supplementary Figure 10a-c**, images for colony forming ability and flow cytometry data following double thymidine block or nocodazole treatment after BAP1 depletion in U-2 OS cells. See the corresponding text in the results section.

Comment 2: “In examining mesothelioma, no hypothesis is stated for the correlative experiments. This results section is the least clear in the paper and the implications for these studies could be elaborated. For example, was there quantitative evidence of ASXL2 destabilization in BAP1 negative/mutant cells. For example, did pathogenic mutants of BAP1 capable of disrupting AXSL1-UBE2E3 axis that correlated with ASXL2 expression? The logistic regression analysis although significant needs some clarification in terms of interpretation”.

Response 2: We thank the reviewer for this comment. Our initial hypothesis is that UBE2Es, ASXL2 and BAP1 act in concert in one major pathway to regulate the DUB activity and function of the BAP1 complex. We expect several layers of regulation including the impact of UBE2Es on the stability of BAP1/ASXL2 complex we characterized here. Thus, cancer-associated deregulated expression and function of UBE2Es, as a consequence of cancer alterations, might result in the loss of BAP1 tumor suppression function. In addition, as BAP1/ASXL2 is a transcription regulatory complex, we also reasoned that inactivating mutations of BAP1 might impact the expression of UBE2Es, as part of feedback regulatory loop (as seen for p53/MDM2 for instance). Therefore, if our hypothesis is correct, we should find statistically significant

correlations among these three proteins (BAP1, ASXL2, and UBE2E3) and we did. Overall, our results suggest that the expression of BAP1, ASXL2 and UBE2Es is correlated further supporting a functional link between these factors. We developed further this section in the revised manuscript (results section). We would like to emphasize that we only performed immunohistochemistry, IHC without studying proteins binding in the tumor specimens used. Therefore, we cannot experimentally conclude about the functional interaction between ASXL2/BAP1 or UBE2E3 in our samples.

Comment 3: “The evidence that pathological BAP1 CTD mutant BAP1 Δ R666-H669 is associated with loss of DEUBAD ubiquitination as well as loss of interaction is of interest. More than half of pathogenic BAP1 mutations map to UCH It would be good to show (perhaps in a graphic) the frequencies of cosmic and germ line single nucleotide variants in relation to CTD and CC1 in BAP1, with potential to disrupt endogenous DEUBAP monoubiquitination. Does the mutation spectra implicate disruption of the DEUBAD-ub ASXL2/UBE3E3 axis universally in cancers such as mesothelioma?”.

Response 3: This is a very interesting point. As we previously reported (Daou et al., J Biol Chem. 2015 Nov 27;290(48):28643-63), several cancer-associated mutations that target critical domains in BAP1 including the UCH catalytic site, the coiled coil 1 (CC1) and the CTD, disrupts ubiquitin binding and BAP1 DUB activity. Also, we are showing in the **Figure 2b-d** that DEUBAD monoubiquitination requires the intramolecular interactions between the UCH, the CC1 and the CTD domains of BAP1 as well as the interaction of the CTD with the DEUBAD of ASXL2 (Mashtalir et al., Mol Cell. 2014; 54(3):392-406; Daou et al., J Biol Chem. 2015; 290(48):28643-63). Therefore, our data suggest that BAP1 cancer-associated mutations within the UCH, CC1 and CTD would disrupt DEUBAD monoubiquitination as the BAP1 Δ R666-H669 mutant does. We added as a **Supplementary Figure 11b**, a graphic describing the type and frequencies of BAP1 cancer mutations occurring within the UCH, CC1 and CTD domains. Importantly, we note that we only selected discrete mutations that result in one or few amino acid changes within a specific domain (Missense mutations and small insertions or deletions). These mutations are extremely useful for evaluating the relative importance of a specific domain. We also emphasize that since the BAP1 NLS is located at the extreme C-terminus, most mutations producing truncations should disrupt the nuclear localization of BAP1 and, as a result, its DUB activity toward histone H2A. We also included graphics showing discrete mutations in important domains of ASXLs (**Supplementary Figure 11c**). In respect to the expression of BAP1, ASXL2 and UBE2E3 in cancer, we analyzed several databases. There are discrepancies between platforms and thus a more comprehensive analysis will be needed to address this aspect in future studies.

Comment 4: “There is phylogenetic conservation of protein partners in the other UCH family protein ie. UCH-L5 which is associated with RPN13 and INO80G which harbour DEUBAD domains. It would be of interest if these are similarly monoubiquitinated as a common mechanism of DUB activation. This would be expected given the highly conserved monoubiquitination of Drosophila ASX.

Response 4: This is an important comment. We have now addressed this concern by conducting similar experiments as we have done for BAP1/ASXLs. We did not observe obvious monoubiquitination signals of INO80G or RPN13 DEUBADs that are dependent on UCH37 expression. In addition we did not identify in INO80G or RPN13 DEUBADs, the conserved lysine found in ASXLs DEUBADs. The data is presented as **Supplementary Figure 2e, f**. See the corresponding text the results section. But further investigations are needed. Perhaps this regulation takes place for UCH37 and its interacting DEUBADs in specific conditions not recapitulated in our assays.

Comment 5: It would be of interest, given the extensive genomic data available publically, to comment on whether or not DEUBAD domain/ASXL2 or UBE2E2 mutations are seen in cancer and whether there is mutually exclusivity with BAP1 SNVs.

Response 5: We have conducted a preliminary analysis. Although ASXL2 and BAP1 mutations were not, in general found in the same cancer patients, we want to be cautious in not stating, at this point, that mutations in the genes encoding these factors are mutually exclusive. In fact, the number of mutations seen in ASXL2 and BAP1 is not enough to draw a definitive conclusion (that is statistically significant). The same applies to UBE2E family of enzymes. It will be interesting to investigate this aspect more carefully in the future. Importantly, it will be highly interesting to also investigate additional mechanisms that can lead to loss of protein expression and function including promoter hypomethylation, microRNA targeting as well as mutations of additional E3 ligases and DUBs that regulate protein stability. Clearly, we are uncovering a major pathway with multiple regulators, and inactivation of one or multiple components of the BAP1-UBE2E-ASXL axis is expected to play an important role in cancer development.

Additional data

- 1) We now show in the **Supplementary Figure 1f** that the DEUBAD domain of ASXL3 is also monoubiquitinated in BAP1-dependent manner.
- 2) We present in the **Supplementary Figure 2e** a sequence alignment of the DEUBAD domains of ASXL2, RPN13 (ADRM1) and INO80G.
- 3) We show in the **Supplementary Figure 2f** that UCH37 do not promote RPN13 or INO80G DEUBADs ubiquitination.
- 4) As in the previous **Supplementary Figure 5d**, we did not show a loading control, beside that we were not be able to find the original blots, we decided to repeat the experiment. We now showing the same result as previously with a blot for tubulin as a loading control.
- 5) We now present as a new **Supplementary Figure 6** the immunostaining of BAP1 DEUBAD (ASXL2), DEUBAD (ASXL2) K370R, ASXL2 WT and ASXL2 K370R for the same experiments presented in **Figure 7 a-c**.

- 6) We added in the **Supplementary Figure 8c**, Caspase-3 staining showing that the overexpression of either Asx WT or Asx K325R in the wing disc do not induce apoptosis.
- 7) We included **Supplementary Figures 10 a-c**, colonies forming and flow cytometry data showing the impact of BAP1 depletion on cell cycle progression.
- 8) We added as **Supplementary Figures 11 b-c**, graphics showing a representation of BAP1, ASXL1, ASXL2 and ASXL3 cancer mutations frequencies within their functional domains.

Reviewer #1 (Remarks to the Author): The authors have addressed all my comments and I would recommend publication of this nice manuscript. Response: We are delighted, no further action needed.

Reviewer #2 (Remarks to the Author): The authors did a very good job of responding to my comments. This is a really interesting paper that deserves publication ASAP. FYI, en-GAL4 is a reporter gene inserted into the invected gene. invected and engrailed are regulated by Polycomb, so it's possible that the dominant negative Asx directly interferes with the transcription of the en-GAL4 driver. No need to mention this in the paper, but it may be a direct target of DUB. Response: We appreciate the comment. We decided to mention in the text that Asx might interfere with en-GAL4 driver. This would help the non-expert reader.

Reviewer #4 (Remarks to the Author): The authors of this revised manuscript have carefully considered the reviewer's comments and these have now been addressed in full with appropriate amendments. Specifically, I am satisfied with the response to my queries from the first review (reviewer 4). In summary, this body of work provides a new and important insight into the regulation of the BAP1's DUB activity. This robust scientific work will be of significant interest to the research community and deserves to be published. Response: We are pleased that the reviewer is satisfied, no further action needed